



# Combining homogeneous and heterogeneous chemistry to model inorganic compounds concentrations in indoor environments: the H$^2$I model (v1.0)

Eve-Agnès Fiorentino[1], Henri Wortham[1], and Karine Sartelet[2]

[1] Laboratory of Environmental Chemistry, CNRS-UMR 7376, Aix-Marseille Université, Marseille, France
[2] CEREA, Joint Laboratory Ecole des Ponts ParisTech - EdF R&D, Université Paris-Est, Champs-sur-Marne, France

**Correspondence:** eve-agnes.fiorentino@univ-amu.fr

**Abstract.** Homogeneous reactivity has been extensively studied over the past years through outdoor air-quality simulations. However, indoor atmospheres are known to be largely influenced by another type of chemistry, that is reactivity with surfaces. Despite progress in the understanding of heterogeneous reactions, these remain barely integrated in numerical models. In this paper, a room-scale indoor air-quality (IAQ) model is developed to represent both heterogeneous and homogeneous
5 chemistry. Thanks to the introduction of sorbed species, deposition and surface reactivity are treated as two separate processes, and desorption reactions are incorporated. The simulated concentrations of inorganic species are compared to experimental measurements acquired in a real room, thus allowing to calibrate the model's undetermined parameters. For the considered experiments durations, the influence of the simulations initial conditions is strong. The model succeeds in simulating correctly the four inorganic species concentrations that were measured, namely NO, NO$_2$, HONO and O$_3$. Each parameter is then varied
10 to estimate its sensitivity and identify the most prevailing processes. The speed of air and building filtration factor are uncertain parameters which appear to have a strong influence, the first one on deposition, and the second one on the control of transport from outdoor. As expected, the NO$_2$ surface hydrolysis plays a key role in the production of secondary species. The secondary production of NO by the reaction of sorbed HONO with sorbed HNO$_3$ stands as an essential component to integrate in IAQ models.

## 1 Introduction

At a time where sustainable development requires more efforts than ever, the improvement of building isolation has become a major concern in the field of construction and renovation. Apart from being necessary for the health and comfort of the occupants, airtight conceptions are needed to curb the energy consumption of accommodations and offices, and thus decrease
20 their carbon footprint. However, as air remains more confined with a lower exchange rate with the outdoor, the pollutants generated indoors have less possibilities to escape, which raises health matters. It is now established that indoor atmospheres





are more polluted than outdoors, while we spend most of our time in indoor environments, up to 90% in developed countries (Carslaw, 2007). In this context, indoor air-quality (IAQ) is bound to be an increasingly studied issue.

Whereas numerical simulations appear as a standard approach for the study of outdoor atmospheres, they are less common in the field of indoor environments. Indoor environments are complex, and processes relying on the surface to volume ratio, which are still not fully understood but often negligible outdoors, get a predominant importance in indoor environments (Weschler, 2011).

Historically, early attempts to model indoor atmospheres focused on the correct assessment of primary emissions, considering each chemical component independently. The model of Nazaroff and Cass (1986) provided the first description of the indoor environment as a chemically reactive homogeneous system, taking into account the interactions of about 30 species and groups of species based on a modified version of the Falls and Seinfeld (1978) mechanism. They included photolytic reactions and a simple form of deposition, considering decomposition and irreversible absorption reactions. Sarwar et al. (2002) adapted the chemical mechanism SAPRC-99 in order to take into account newer advances on $O_3$/alkenes reactions. In particular, they introduced the chemistry of 40 VOCs recognized as atmospheric pollutants. Deposition was modelled as in Nazaroff and Cass (1986), and no deposition was assumed for species for which no deposition velocity was available. A more detailed chemical mechanism was tested by Carslaw (2007), who adapted the Master Chemical Mechanism (MCM) to indoor environments, including about 4600 species and 15400 reactions. Deposition was modelled similarly to Nazaroff and Cass (1986), and for the first time, a heterogeneous reaction on indoor surfaces was considered, by introducing a production rate accounting for HONO secondary formation. Later, Mendez et al. (2015) implemented a simplified version of the SAPRC-07 mechanism and described deposition as a two-step process, by making a distinction between transport from free-space to surface, and reactivity with the surface. Further details were provided by Mendez et al. (2017), who parameterized the mass transfer effect using a model of transport-limited deposition velocity.

As underlined by Weschler (2011), surface chemistry is responsible for secondary pollutant formation which can be of greater importance for IAQ than the primary emissions. Because heterogeneous reactions can be faster than their equivalent gas-phase homogeneous reaction, their importance relative to the air exchange rate and thus their influence on indoor atmospheres is large. Secondary pollutants can persist a long time after the reagent species have come to negligible levels, and are very difficult to anticipate, due to their strong dependence on ambiant conditions.

The heterogeneous hydrolysis of $NO_2$ is one of these reactions, and is recognized as the main source of HONO in indoor environments. There is evidence that certain surfaces can act as a reservoir of sorbed $NO_2$, and that these surfaces can release HONO long after the decay of $NO_2$ (Wainman et al., 2001). Presumably, this HONO surface release depends on the ambiant $NO_2$ concentration, ambiant relative humidity and on the surface ability to retain water.

As a rule, it is assumed that the heterogeneous hydrolysis of $NO_2$ leads to the formation of HONO and $HNO_3$ following the stoichiometry proposed by Febo and Perrino (1991) :

$$2\,NO_2 + H_2O \rightarrow HONO + HNO_3. \tag{R1}$$





Contrary to HONO, $HNO_3$ is not observed as gas-phase product in this process, due to its strong adsorption capacity. $HNO_3$ remains on the surface and can react with other species. Namely, Mochida and Finlayson-Pitts (2000) studied the production of $NO_2$ by the reaction of $HNO_3$ with gaseous NO on porous glass. They showed that the NO concentration cannot decay below a threshold value, suggesting NO regeneration pathways. Coherently, NO formation was pointed out during $NO_2$ hydrolysis experiments, simultaneously to the $NO_2$ exposure or at longer reaction times. Finlayson-Pitts et al. (2003) measured for this

reaction a HONO yield that was much less than 50% of the $NO_2$ loss, and observed that the yield of NO relative to HONO increased with time. Based on their own and previously reported observations, they suggested NO could be formed by the secondary autoionization of the sorbed HONO such as

$$2\,HONO \rightarrow NO + NO_2 + H_2O \tag{R2}$$

and also by conversion of the sorbed HONO following a mechanism that involves $HNO_3$ and simplifies to the net reaction

$$HONO + HNO_3 \rightarrow 2\,NO + H_2O + O_2. \tag{R3}$$

Considering longer reaction times, NO could also react with $HNO_3$ following the reaction (Finlayson-Pitts et al., 2003)

$$NO + HNO_3 \rightarrow HONO + NO_2. \tag{R4}$$

Finally, the photochemical enhancement of the $NO_2$ hydrolysis was investigated by Ramazan et al. (2004), who showed that the $NO_2$ hydrolysis was not photo-enhanced itself, but that HONO production was fostered by another process, which was

likely the photolysis of sorbed $HNO_3$ caused by the UV rays. Other heterogenous reactions could be pointed out, especially those involving $O_3$ which is known to have a significant adsorption capacity.

Reviewing the state of the art of surface reactions unveils a serious void in the modelling of indoor atmospheres. Current models incorporate these phenomena quite incompletely, due to the strong uncertainties they are subjected to. In particular, the ratio of production of NO compared to HONO during the $NO_2$ hydrolysis derives from mechanisms that are still unclear.

The detailed scheme proposed by Finlayson-Pitts et al. (2003) involves the reactions of several intermediate species whose reaction rates are unknown. As a test, they introduced this scheme in the kinetic program REACT to compute the loss of $NO_2$ and formation of HONO and NO, and adjusted the required rate constants to obtain a good match with their cell experiments data. In this model, uptake and reactions on surfaces were not explicitly treated, making heterogeneous reactions implemented like gas-phase reactions. Ramazan et al. (2004) made a similar work, and the parameterization they proposed was later used by

Courtey et al. (2009) to model confined atmospheres, *i.e.* without including ventilation and primary emissions.

In this work, a room-scale IAQ model is developed, to incorporate the above described heterogeneous chemistry, in addition to the homogeneous chemistry already considered by box models. The concentrations simulated by this model are compared to measurements that were performed in a real office (Gandolfo, 2018; Gandolfo et al., in prep) from the $27^{th}$ to the $31^{st}$ October 2016 in Martigues (France). The aim of these measurements was to study the impact of photocatalytic paints characterized in

laboratory experiments (Gandolfo et al., 2015, 2017) on indoor atmospheres healthiness. Whereas laboratory tests were conducted with paints containing up to 7% of $TiO_2$ nanoparticles, this campaign was restricted to the use of a non-photocatalytic





paint (reference paint), and to the same paint enriched with a commercially viable amount of 3.5% of $TiO_2$ nanoparticles. Two types of measurements were obtained, with UVs-blocking and borosilicate windows. The simulations presented in this paper are compared to the data obtained with the UVs-blocking window only, so as to minimize the effect of photo-induced

processes, which will be studied in a separate paper. The organic compounds concentrations are fixed to their measured values, so as to focus on the modelling of inorganic species.

The present model, called $H^2I$ (Homogeneous Heterogeneous Indoor) model, assumes a uniformly mixed environment, taking into account emissions, ventilation, chemistry and deposition processes. The chemical mechanism solving the gas-phase chemistry is RACM2 (Goliff et al., 2013). Deposition is modelled as in Mendez et al. (2017). As in Finlayson-Pitts et al.

(2003) and Ramazan et al. (2004), the rate constants of the heterogeneous reactions are adjusted so as to obtain a reasonable match with the experimental data. Contrary to other box models (Nazaroff and Cass, 1986; Sarwar et al., 2002; Carslaw, 2007; Courtey et al., 2009; Mendez et al., 2015), this model does not assume the light to be homogeneous throughout the room. Here, two different parts are considered, one irradiated by direct light and another one illuminated indirectly. It is the first two-box model allowing to consider the evolution of light intensity of each part along the day, as well as the volumes they occupy (Tlili

et al., in prep).

First, this paper gives a detailed description of the $H^2I$ model. The input data and the tunable parameters of the model are then described. These parameters are calibrated by comparing the simulation results to the concentration profiles which were acquired in Martigues. For each experiment, the set of parameters leading to the best simulation, called reference simulation, is given. Each of the parameters is then varied so as to estimate its sensitivity, and identifying the most impacting processes.

Finally a surface saturation limit is implemented to test the model in high $NO_2$ conditions.

## 2   Presentation of the $H^2I$ (Homogeneous Heterogenous Indoor) model

### 2.1   Master equation

The room is divided into two volumes, a volume illuminated by the light of the sun, and a darker one illuminated by indirect light. As these lights have different intensities, the magnitudes of the photolytic reactions occurring inside these volumes are

different, leading to different concentrations in each volume. $m_i^j$ [μg] is the mass of species $i$ in the box $j$, with $j = \{L, S\}$, $L$ denoting the box illuminated by direct light and $S$ denoting the shaded box. The evolution of $m_i^j$ with time is given by the classical box model equation (e.g. Sarwar et al. (2002); Carslaw (2007)) complemented by a box exchange term (Furtaw Jr. et al., 1996) :

$$\frac{\mathrm{d}m_i^j}{\mathrm{d}t} = k_{\mathrm{AER}}f m_i^{\mathrm{Out}} - k_{\mathrm{AER}}m_i^j - k_{\mathrm{DEP},i}^j m_i^j + k_{\mathrm{BOX}}^j \Delta_j m_i + \sum_p Q_{p,i}^j + \sum_q R_{i,q}^j \tag{1}$$

where $t$ is the time [s], $k_{\mathrm{AER}}$ is the air exchange rate between the room and its outside [s$^{-1}$], including the rest of the building, $f$ is the outdoor-to-indoor filtration factor [-], *i.e.* the fraction of air exchange with outdoor, $m_i^{\mathrm{Out}}$ is the outdoor mass of species $i$ introduced in box $j$ [μg], $k_{\mathrm{DEP},i}^j$ is the deposition rate of species $i$ [s$^{-1}$], $k_{\mathrm{BOX}}^j$ is the air exchange rate between the boxes [s$^{-1}$],





$\Delta_j m_i$ is the corresponding mass transfer [$\mu$g], $Q_{p,i}$ is the emission rate of source $p$ [$\mu$g.s$^{-1}$] and $R_{i,q}^j$ is the mass reaction rate between species $i$ and species $q$ [$\mu$g.s$^{-1}$].

By denoting $V_{\text{box}}^j$ the volume of the box $j$ [m$^3$], the mass transfer from box $L$ to box $S$ is expressed as

$$\Delta_L m_i = -m_i^L + \frac{V_{\text{box}}^L}{V_{\text{box}}^S} m_i^S \tag{2}$$

and the mass transfer from box $S$ to box $L$ as

$$\Delta_S m_i = \frac{V_{\text{box}}^S}{V_{\text{box}}^L} m_i^L - m_i^S. \tag{3}$$

The evolution of the species concentrations is obtained by dividing Eq. (1) by the box volume. This yields

$$\frac{dC_i^j}{dt} = k_{\text{AER}} f C_i^{\text{Out}} - k_{\text{AER}} C_i^j - k_{\text{DEP},i}^j C_i^j + k_{\text{BOX}}^j \Delta_j C_i + \sum_p \frac{Q_{pi}^j}{V_{\text{box}}^j} + \sum_q \frac{R_{iq}^j}{V_{\text{box}}^j} \tag{4}$$

where $C_i^j = m_i^j / V_{\text{box}}^j$ is the indoor concentration of species $i$ in volume $j$ [$\mu$g.m$^{-3}$], $C_i^{\text{Out}}$ is the outdoor concentration of species $i$ [$\mu$g.m$^{-3}$] and $\Delta_j C_i$ is the concentration variation caused by the gas transfers between boxes [$\mu$g.m$^{-3}$], given by

$$\begin{aligned} \Delta_L C_i &= -C_i^L + C_i^S, \\ \Delta_S C_i &= C_i^L - C_i^S. \end{aligned} \tag{5}$$

On the right-hand side of Eq. (4), the first term is the intake of species coming from outdoors. The second term is the
concentration loss due to the leakages toward outdoors, but also toward the other rooms of the building. The third term is deposition. The fourth term brings in the mixing between the two volumes. The fifth term represents the indoor sources that release species $i$ and the last term is the contribution of the reactions involving species $i$.

The two types of sources encountered in these experiments are the emissions of the paint boards placed on the walls, $Q_{\text{paint},i}$, and the emissions of the building itself, $Q_{\text{room},i}$. The room emissions in the box $j$ can be written as

$$Q_{\text{room},i}^j = Q_{\text{room},i} \frac{V_{\text{box}}^j}{V_{\text{room}}}. \tag{6}$$

The paint emissions are derived from their surface emission rates :

$$Q_{\text{paint},i}^j = E_{\text{paint},i}^j S_{\text{paint}}^j \tag{7}$$

where $E_{\text{paint},i}$ are the paint surface emission rates obtained experimentally [$\mu$g.m$^{-2}$.s$^{-1}$] and $S_{\text{paint}}^j$ is the surface of paint in the box $j$ [m$^{-2}$]. In the box illuminated by direct light, Eq. (4) thus gives

$$\frac{dC_i^L}{dt} = k_{\text{AER}}(f C_i^{\text{Out}} - C_i^L) - k_{\text{DEP},i}^L C_i^L + k_{\text{BOX}}^L(-C_i^L + C_i^S) + \frac{Q_{\text{room},i}}{V_{\text{room}}} + \frac{E_{\text{paint},i}^L S_{\text{paint},i}^L}{V_{\text{box}}^L} + \sum_q \frac{R_{iq}^j}{V_{\text{box}}^L} \tag{8}$$

and in the shaded box, Eq. (4) gives

$$\frac{dC_i^S}{dt} = k_{\text{AER}}(f C_i^{\text{Out}} - C_i^S) - k_{\text{DEP},i}^S C_i^S + k_{\text{BOX}}^S(C_i^L - C_i^S) + \frac{Q_{\text{room},i}}{V_{\text{room}}} + \frac{E_{\text{paint},i}^S S_{\text{paint},i}^S}{V_{\text{box}}^S} + \sum_q \frac{R_{iq}^j}{V_{\text{box}}^S}. \tag{9}$$





## 2.2 Boxes evolution and exchanges between the boxes

We denote $V_{\text{box}}^L$ and $V_{\text{box}}^S$ the volumes of the sunlit and shaded boxes. Accordingly, the total surface of the room $S_{\text{room}}$ is split

into two parts, $S_{\text{box}}^L$ and $S_{\text{box}}^S$. Their evolutions along the day are constrained by the relationships

$$V_{\text{room}} = V_{\text{box}}^L + V_{\text{box}}^S$$
$$S_{\text{room}} = S_{\text{box}}^L + S_{\text{box}}^S$$

(10)

where $V_{\text{room}} = 32.8\,\text{m}^3$ is the total volume of the room and $S_{\text{room}} = 62.7\,\text{m}^2$.

Hourly values of $V_{\text{box}}^L$ and $S_{\text{box}}^L$ were estimated by modelling the solar flux in the room (Tlili et al., in prep). The position

of the sun and extrapolation of its beams from the windows were assessed with the Revit 2018 software; the irradiated surface

and beams volume were then calculated by vertical and horizontal projections of the indoor solar flux using Autocad 2016 (see

www.autodesk.fr for both softwares). The evolution of $V_{\text{box}}^L$ and $S_{\text{box}}^L$ as a function of time $t_h$ [h] is inferred from these values

by fitting a Gaussian law (Fig. 1) :

$$V_{\text{box}}^L(t_h) = \frac{A_v}{\sigma_v \sqrt{2\pi}} e^{-\frac{(t_h - \mu_v)^2}{2\sigma_v^2}}$$

(11)

with $A_v = 36.505\,\text{m}^3.\text{h}$, $\sigma_v = 2.154\,\text{h}$ and $\mu_v = 11.199\,\text{h}$,

$$S_{\text{box}}^L(t_h) = \frac{A_b}{\sigma_b \sqrt{2\pi}} e^{-\frac{(t_h - \mu_b)^2}{2\sigma_b^2}}$$

(12)

with $A_b = 36.958\,\text{m}^2.\text{h}$, $\sigma_b = 2.1950\,\text{h}$ and $\mu_b = 11.555\,\text{h}$. $V_{\text{box}}^S$ and $S_{\text{box}}^S$ are deduced from $V_{\text{box}}^L$ and $S_{\text{box}}^L$ using Eq. (10).

$S_{\text{box}}^L$ and $S_{\text{box}}^S$ divide the total solid surface of the experimental room, including walls, window, floor and ceiling. The

complement of $S_{\text{box}}^L$ to obtain the surface of the volume $V_{\text{box}}^L$ is the same as the complement of $S_{\text{box}}^S$, necessary to obtain

the surface of $V_{\text{box}}^S$. This complement is the surface allowing gas transfer between the boxes, and is denoted $S_{\text{gas}}$. This latter

was estimated with the same method as the one used for $V_{\text{box}}^L$ and $S_{\text{box}}^L$ (Tlili et al., in prep), giving (Fig. 1)

$$S_{\text{gas}}(t_h) = \frac{A_g}{\sigma_g \sqrt{2\pi}} e^{-\frac{(t_h - \mu_g)^2}{2\sigma_g^2}}$$

(13)

where $A_g = 120.04\,\text{m}^2.\text{h}$, $\sigma_g = 2.4683\,\text{h}$ and $\mu_g = 11.154\,\text{h}$.

The variation of mass within the boxes due to the air mixing is proportional to the surface $S_{\text{gas}}$. This proportionality is

expressed by the air exchange constant $k_{\text{BOX}}$, defined as (Furtaw Jr. et al., 1996)

$$k_{\text{BOX}}^j = u_{\text{inf}} \frac{S_{\text{gas}}}{V_{\text{box}}^j}$$

(14)

where $u_{\text{inf}}$ is the average speed of air in the room. This velocity was estimated based on a tracer injection experiment and

numerical tests (see sections 3.2 and 5.1.1).





## 2.3 Chemical mechanism

Numerical simulations cannot afford the computation of the millions of reactions occurring in real atmospheres. Approxima-
tions are required to reduce this complexity and alleviate computational efforts. Modellers can opt for different types of kinetic
chemical mechanisms, depending on the targeted accuracy. In particular, the lumped-species approach allows to make use of
a reduced number of compounds, each compound representing several species having similar properties (Gery et al., 1989;
Stockwell et al., 1990; Yarwood et al., 2005; Carter, 2010; Goliff et al., 2013), such as reactivity with OH or carbon bounds.
A given species can be represented by a single model compound, or by the combination of several model compounds. Mendez
et al. (2015) compared the concentrations they obtained with such kind of lumped-species model, SAPRC-07 (Carter, 2010),
to the concentrations Carslaw (2007) simulated with the detailed chemistry model MCM, and concluded that their overall
behaviors were consistent, with respect to the $O_3$, NOx (NO + $NO_2$) and HOx (HO + $HO_2$) variations. Considering that the
general dynamics of homogeneous indoor chemistry can be reproduced by semi-explicit models initially developed for out-
door atmospheres, RACM2 (Goliff et al., 2013), which is also a lumped-species based model, is used in this paper to solve the
chemical reactivity.

To introduce the surface chemistry highlighted by laboratory studies but hardly present in current indoor models, the RACM2
scheme is modified so as to take into account the heterogenous reactions listed in Tab. 1. The resulting modified version of the
RACM2 scheme comprises 117 species and 389 reactions among which 34 are photolytic and 38 heterogeneous. To investigate
further the reactions highlighted in the introduction, some surface species are introduced, namely $NO_{(ad)}$, $NO_{2\,(ad)}$, $HONO_{(ad)}$
and $HNO_{3\,(ad)}$. These species can either sorb, desorb or react together. $HNO_{3\,(ad)}$ is not allowed to desorb, to account for the ex-
perimental observation that the $NO_2$ hydrolysis never yields gaseous $HNO_3$ (Finlayson-Pitts et al., 2003). The kinetic constants
of desorption and surface reactions are uncertain, and thus considered as tunable parameters. Adsorption and decomposition
reactions are modelled by combining transport to the boundary layer and surface adhesion (Mendez et al., 2015), as now
detailed.

## 2.4 Deposition and surface reactivity

This section details the computation of the kinetic constants of the adsorption and decomposition reactions. The local deposition
rate $k_{\mathrm{DEP},i}^{j}$ is modelled as the equivalent of two resistances in parallel, one corresponding to the transport-limited deposition
rate $k_{\mathrm{tran},i}^{j}$ and one corresponding to the surface adhesion rate $k_{\mathrm{react},i}^{j}$ :

$$\frac{1}{k_{\mathrm{DEP},i}^{j}} = \frac{1}{k_{\mathrm{tran},i}^{j}} + \frac{1}{k_{\mathrm{react},i}^{j}}. \tag{15}$$

When $k_{\mathrm{react},i}^{j}$ is larger than $k_{\mathrm{tran},i}^{j}$ the species loss is limited by the transport to the surface boundary layer. A contrario, when
$k_{\mathrm{tran},i}^{j}$ is larger than $k_{\mathrm{react},i}^{j}$, species removal is limited by surface reactivity (Grøntoft and Raychaudhuri, 2004).





### 2.4.1 Transport-limited deposition rate

The rate constant $k^j_{\text{tran},i}$ can be expressed as

$$k^j_{\text{tran},i} = v_{\text{trd},i} \frac{S^j_{\text{box}}}{V^j_{\text{box}}} \tag{16}$$

where $v_{\text{trd},i}$ is the deposition velocity limited by transport. It is computed using the method of Lai and Nazaroff (2000), following the same approach as Mendez et al. (2017) to model the heterogeneous production of HONO :

$$v_{\text{trd},i} = v^{ad}_{\text{trd},i} u^* \tag{17}$$

where $v^{ad}_{\text{trd},i}$ is a dimensionless deposition velocity whose computation is detailed below, and $u^*$ is the friction velocity defined by

$$u^* = \left( \nu \left. \frac{dU}{dy} \right|_{y=0} \right)^{1/2} \tag{18}$$

with $U$ the mean air speed parallel to the surface [m.s$^{-1}$], $y$ the distance from the surface [m] and $\nu$ the air kinematic viscosity [m$^2$.s$^{-1}$], defined by $\nu = \eta/\rho$ with $\eta$ the air dynamic viscosity [kg.m$^{-1}$.s$^{-1}$] and $\rho$ the air volumetric mass [kg.m$^{-3}$]. Considering the narrow temperature range encountered in these experiments, $\rho$ is approximated with the ideal gas law. The viscosity $\eta$ is expressed as function of temperature $T$ [K] using the semi-empirical Sutherland relationship

$$\eta(T) = \eta_0 \left( \frac{T}{T_0} \right)^{3/2} \frac{T_0 + S}{T + S} \tag{19}$$

where the Sutherland's constant for air is $S = 113$ (Kaper and Ferziger, 1972), and $\eta_0 = 1.783 \times 10^{-5}$ kg.m$^{-1}$.s$^{-1}$ at $T_0 = 288.15$ K.

The derivative of $U$ is given by

$$\left. \frac{dU}{dy} \right|_{y=0} = \left( \frac{0.074}{\rho\nu} \right) \left( \frac{\rho u^2_{\text{inf}}}{2} \right) \left( \frac{u_{\text{inf}} L}{\nu} \right)^{-1/5} \tag{20}$$

where $L$ is a characteristic length of the room surface, typically $L = (V_{\text{room}})^{1/3}$.

As in Mendez et al. (2017), gravity is assumed to be negligible for gases, *i.e.* the dimensionless deposition velocity $v^{ad}_{\text{trd},i}$ is the same for horizontal and vertical surfaces. Assuming that the molecules eddy diffusivity equals the fluid turbulent viscosity $\nu_t$ in indoor environments, Lai and Nazaroff (2000) showed that

$$\frac{1}{v^{ad}_{\text{trd},i}} = \int_{r_0}^{30} \left( \frac{1}{\frac{\nu_t}{\nu} + \frac{D_i}{\nu}} \right) dy^{ad} \tag{21}$$

where $D_i$ is the diffusion coefficient of species $i$ [m.s$^{-1}$], $y^{ad}$ is the adimensional distance from the surface, and $r_0$ is the radius of the particle, taken here as zero. The ratio $\nu_t/\nu$ is given by Lai and Nazaroff (2000) for several intervals of $y^{ad}$. The





diffusion coefficient can be estimated based on the species critical temperature $T_c$ [K] and critical volume $V_c$ [cm$^3$.mol$^{-1}$], following various models. Among all the models tested in the comparative study of Ravindran et al. (1979), the model of Chen and Othmer (1962) is the one that showed the best agreement with their experimental data. Considering a species $i$ diffusing in air, this model gives

$$D_i = \frac{4.3 \times 10^{-5} \left(\frac{T}{100}\right)^{1.81} [(M_{air} + M_i)/(M_{air}M_i)]^{1/2}}{P \left(\frac{T_{c,air}T_{c,i}}{10^4}\right)^{0.1406} \left[\left(\frac{V_{c,air}}{100}\right)^{0.4} + \left(\frac{V_{c,i}}{100}\right)^{0.4}\right]^2} \tag{22}$$

where $P$ is the ambient pressure [atm], $M_i$ is the species molar mass and $M_{air} = 28.97$ g.mol$^{-1}$.

The diffusion coefficients are computed with this method for each RACM2 compound listed in Tab. 2 with the parameters used for this calculation. In this table, $T_c$ and $V_c$ are the critical temperatures and volumes of chemical species that are considered to be representative of the RACM2 lumped-compounds. The references from which the $T_c$ values are taken are specified. When there is no reference, $T_c$ is computed with the method of Joback and Reid (1987). As experimental values of $V_c$ are difficult to find for a variety of species, they are also computed with the method of Joback and Reid (1987), excepted for methanol, xylene, and butanol, whose reference provides both $T_c$ and $V_c$.

### 2.4.2 Surface reaction rate

Surface adhesion is modelled with the rate constants $k_{\text{react},i}^j$ which are defined as

$$k_{\text{react},i}^j = \frac{\gamma_i \omega_i}{4} \frac{S_{\text{box}}^j}{V_{\text{box}}^j}, \tag{23}$$

where $\gamma_i$ is the uptake coefficient [-] and $\omega_i$ the thermal velocity [m.s$^{-1}$] of species $i$. $\gamma_i$ is the ratio of collisions of species $i$ with the surface that yield reaction or simple deposition, to the total number of collisions. $\omega_i$ depends on the the temperature and can be expressed as

$$\omega_i = \sqrt{2.1171 \times 10^4 \frac{T}{M_i}}. \tag{24}$$

The uptake coefficient is characteristic of the relationship between the species and the surface itself. It has been determined experimentally with the paints used in this study for two species, NO$_2$ and xylene, meaning that uptake coefficients of other species are unknown. The deposition of organic species is out of the scope of this paper, as organic concentrations are set to the measured values here. The uptake coefficients of NO, HONO and O$_3$ are uncertain, and thus considered as tunable parameters. Since the simulated concentrations of NO$_3$, HNO$_3$, HNO$_4$ and H$_2$O$_2$ are low, they are given an infinite uptake for simplicity, so that their deposition is only controlled by transport ($k_{\text{DEP},i}^j = k_{\text{tran},i}^j$). By default, the same procedure is applied to the HO$_2$ radical, noting that when its deposition is neglected, the resulting difference in average NOx concentration is of the order of $0.1 \mu$g.m$^{-3}$. The deposition of the HO radical is considered as negligible compared to its consumption by reactivity (Sarwar et al., 2002).





### 2.4.3 Parameterization of $\gamma_{NO_2}$

The uptake coefficient $\gamma_{NO_2}$ was measured in various laboratory conditions by Gandolfo et al. (2015, 2017). This section details how parameterizations are inferred from these measurements and how they are used to calculate $\gamma_{NO_2}$ as a fonction of ambiant conditions.

The measurements made as a function of the relative humidity, denoted $H$, are normalized by the measurement made at $H_{\text{ref}}$ = 40%. This gives, using a polynomial fitting :

$$\gamma_{NO_2}^{\text{norm},H}(H) = \sum_{k=0}^{2} a_k H^k \tag{25}$$

where $a_0 = 0.706$, $a_1 = 1.50 \times 10^{-2}$ and $a_2 = -2.31 \times 10^{-4}$.

Considering that $\gamma_{NO_2}$ varies with the $NO_2$ concentration in the room, measurements were made as a function of the $NO_2$ concentration in ppb, denoted $N$. By normalizing these measurements by the measurement made at $N_{\text{ref}} = 40$ ppb, an exponential fitting gives :

$$\gamma_{NO_2}^{\text{norm},N}(N) = 118.06 \exp^{\frac{-(N+64.52)}{20.41}} + 0.61. \tag{26}$$

Measurements were also made as a function of the light intensity irradiating the surface, denoted $I$. The light intensity produced by the reactor covered a spectrum ranging from 340 nm to 400 nm. Because the paint photocatalytic effect reaches saturation above a certain light threshold, a function type that does not increase too much at high intensity is chosen to express $\gamma_{NO_2}^{\text{norm},I}$. Using the measurements normalized by the measurement made at $I_{\text{ref}} = 8.5$ W.cm$^{-2}$, a logarithmic fitting gives :

$$\gamma_{NO_2}^{\text{norm},I}(I) = \ln(I + 19.63) - 2.74. \tag{27}$$

Contrary to the other measurements, the measurements made as a function of temperature were performed at $I = 20$ W.m$^{-2}$. By dividing the measurements as a function of $I$ by the measurement made at $I = 20$ W.m$^{-2}$, a relationship similar to Eq. (27) is obtained, and denoted $\gamma_{NO_2}^{\text{norm},I_2}$. By multiplying the measurements as a function of temperature by $\gamma_{NO_2}^{\text{norm},I_2}(I_{\text{ref}})$, these measurements are brought to the same conditions of irradiance as the other measurements. They are then divided by the measurement made at $T_{\text{ref}} = 296$ K. Finally, a polynomial fitting gives :

$$\gamma_{NO_2}^{\text{norm},T}(T) = b_1 T_s(T) + b_0 \tag{28}$$

where $b_0 = -17.62$ and $b_1 = 6.25 \times 10^{-2}$ K$^{-1}$ for the paint containing 3.5 % of $TiO_2$. The values $b_0 = 1$ and $b_1 = 0$ are preferred for the reference paint, considering that the observed decreasing trend falls within the measurements uncertainty. $T_s$ is the temperature of the surface of paint [K]. In a real room, $T_s$ depends on a variety of factors, including location, season, orientation, ambiant temperature and surface coating (Shen et al., 2011). For the simulations, $T_s$ is set such as $T_s = T$ in the shaded box and $T_s = 1.2 \times (T - 273.15) + 273.15$ in the sunlit box.

The $\gamma_{NO_2}$ at a given set of parameters $H$, $N$ and $T$ is calculated from the reference uptake coefficient $\gamma_{NO_2}^{\text{ref}}$ measured at $T_{\text{ref}}$ = 296 K, $H_{\text{ref}} = 40$%, $I_{\text{ref}} = 8.5$ W/m$^{-2}$ and $N_{\text{ref}} = 40$ ppb, according to

$$\gamma_{NO_2}(H,N,T) = \gamma_{NO_2}^{\text{ref}} \gamma_{NO_2}^{\text{norm},N}(N) \gamma_{NO_2}^{\text{norm},H}(H) \gamma_{NO_2}^{\text{norm},T}(T). \tag{29}$$





The parameterization $\gamma_{NO_2}^{norm,I}$ is not considered in Eq. (29), given the fact that it was established based on measurements with a light spectrum ranging from 340 nm to 400 nm, and that the measurements presented in this paper were obtained with a light spectrum starting from 395 nm.

The dependence of $\gamma_{NO_2}^{ref}$ with the percentage of TiO$_2$ nanoparticles contained by the paint and the parameterizations of 285 $\gamma_{NO_2}^{norm}$ are presented in Fig. 2. $\gamma_{NO_2}^{ref}$ increases with % TiO$_2$ but the values at 0 % and 3.5 % are very close. In the simulations, $\gamma_{NO_2}^{ref} = 5 \times 10^{-6}$ is used for both paints and for the surface that is not covered by paint (floor, ceiling, rest of the walls).

## 2.5  Photolysis

In all of the indoor models presented in the introduction, light was assumed to have two origins, sunlight and artificial light. Using the indoor light intensity recommended for reading purposes, Sarwar et al. (2002) assumed that each light source ac-290 counted for 50% of the total light, and combined accordingly spectral power distributions obtained from the literature to obtain the total spectral distribution. Nazaroff and Cass (1986), Mendez et al. (2015) and Carslaw (2007) computed their own outdoor photon fluxes, and applied attenuation factors to account for window filtration. Carslaw (2007) and Mendez et al. (2015) used the same indoor light fluxes as Nazaroff and Cass (1986), and started with the same attenuation factors before varying them.

Whereas light intensity is homogeneous in the direct light whatever the distance from the window, it decreases rapidly as 295 getting away from the direct sunlight (Gandolfo et al., 2016). The distribution of light intensity in the shaded volume is strongly location-specific, and thus hard to predict. However, light intensity in the shaded volume is much lower than the intensity of direct light, so the impact of the photolytic reactions occurring in the shaded volume is minor compared to those occurring in the sunlit volume. As an approximation, the photolysis constants in the shaded box are computed using a unique actinic flux which was measured close to the area illuminated by the sunlight. This model does not represent the light decrease as getting 300 away from the window, because only two boxes are considered, shaded and sunlit.

Let $\zeta$ be the indoor actinic flux [photons.cm$^{-2}$.s$^{-1}$.nm$^{-1}$] measured at $t = t_{ref}$. Let $\lambda_{min}$ and $\lambda_{max}$ be the minimum and maximum wavelengths of the light spectrum [nm]. The photolysis constants associated to this actinic flux are given by (Nazaroff and Cass, 1986) :

$$J_i^{ref} = \int\limits_{\lambda_{min}}^{\lambda_{max}} \zeta(\lambda)\kappa_i(\lambda)\phi_i(\lambda)\mathrm{d}\lambda \tag{30}$$

where $J_i^{ref}$ is the photolysis constant of species $i$ [photons.cm$^{-2}$.s$^{-1}$] at $t = t_{ref}$, $\kappa_i$ the species cross section [-] and $\phi_i$ the species quantum yield [-]. The actinic flux used to calculate $J_i^{ref}$ in the indirect light was measured at $t_{ref} = 11$ h (GMT) on the 27$^{th}$ October (Experiment 1), and the one used to calculate $J_i^{ref}$ in the direct light was measured at $t_{ref} = 11$ h (GMT) on the 29$^{th}$ October (Experiment 3). The light spectrum starts at $\lambda_{min} = 390$ nm in the direct light, and $\lambda_{min} = 394$ nm in the indirect light. Both spectra end at 660 nm.

To account for the evolution of the photolytic constants with day time, a parameterization is inferred from the HONO, NO$_2$, HCHO, H$_2$O$_2$ and NO$_3$ photolysis rates measured by a spectroradiometer in the direct light on the 30$^{th}$ October, with windows



that did not cut UV rays (Fig. 3) :

$$J_i(\theta) = A_i \exp^{\frac{-(\theta - C)}{B}}, \tag{31}$$

where $\theta$ is the zenith angle and $J_i$ is the photolysis constant of species $i$ as a function of $\theta$. The evolution of $\theta$ with day hour is

presented in Fig. 4. The curves fitting the $J_i$ rates measured in the morning and those fitting the ones measured in the afternoon are superimposed in Fig. 3, indicating no hysteresis. For each compound, the values of $B$ and $C$ are very close, with an average of $B = 10.77$ and $C = 50.03$. For a given compound, the prefactor $A$ is given by

$$A_i = J_i^{\mathrm{ref}} \exp^{\frac{\theta_{\mathrm{ref}} - C}{B}}, \tag{32}$$

where $\theta_{\mathrm{ref}}$ is the zenith angle corresponding to $t_{\mathrm{ref}}$. This yields

$$J_i(\theta) = J_i^{\mathrm{ref}} \exp^{\frac{-(\theta - \theta_{\mathrm{ref}})}{B}}. \tag{33}$$

Eq. (33) is plotted in Fig. 3 using the $J_i^{\mathrm{ref}}$ calculated with the $\phi$ and $\kappa$ values available for the RACM2 chemical mechanism (Kim et al., 2009) in the Polyphemus air-quality modelling platform. We observe a reasonable agreement between the measured and calculated photolysis rates.

## 2.6    Numerical resolution

The simulations start and end at the times fixed by the user (see section 3.1). The time step $\Delta t$ of the main loop of time of the program is set to $100\,\mathrm{s}$. It corresponds to an input/output time step : at the beginning of each iteration of the main loop, input data such as temperature, humidity, and outdoor concentrations are read from a file, and at the end of each iteration, the concentrations are written to a file. Variables characterizing the environment, source and sink terms are also initialized and updated in the main loop, namely box volumes and surfaces (Eqs. 10-12), box air exchange (Eq. 14), ventilation, supply from

outdoors, and emissions.

The resolution of Eq. (4) is performed using operator splitting : the evolution of the concentrations due to emissions, air exchanges between boxes and between the room and the outside is first solved using the explicit trapezoidal rule (ETR), which is an explicit second order solver corresponding to a two-stage Runge-Kutta method (Ascher and Petzold, 1998). The time step is adapted as described in Sartelet et al. (2006) : each main time step $\Delta t$ of $100\,\mathrm{s}$ is decomposed in sub-time steps $\delta t_k$

determined by the ETR method, such as $\Delta t = \sum_k \delta t_k$. After each sub-time step $\delta t_k$, the third and last terms of the right-hand side of Eq. (4) are solved together. The evolution of the concentrations due to homogeneous and heterogeneous reactions and deposition is computed using the Rosenbrock 2 (ROS2) algorithm (Rosenbrock, 1963; Sandu et al., 1997), with time steps automatically adapted between $\delta t_k$ and $\delta t_{k+1}$ by the ROS2 algorithm.

In this paper, the VOCs are assigned to their experimental values at each iteration of the main loop. By imposing the

VOCs concentrations at each main iteration, no drift is observed between experimental and calculated values, meaning that the characteristic time of their evolution caused by chemical reactivity, sources and sinks is slower than the model time step $\Delta t$. Note that this is not the case with more reactive species, such as NO and $NO_2$, which may evolve significantly between two iterations.





## 3 Input data and parameters for model evaluation

### 3.1 Measurements used as input and model/measurement comparisons

Three experiments were conducted with anti-UVs windows. The first one was conducted without any paint board ("naked" walls), the second one with walls covered by the reference paint, and the third one with walls covered by the 3.5% $TiO_2$ paint. The complete description of these experiments is provided by Gandolfo et al. (in prep) (see also Gandolfo (2018)). The current section provides a brief introduction of the data used in this paper.

The room was ventilated before each experiment. The start time of the simulation is chosen so as to match with the beginning of the VOC concentrations rise caused by the windows closing. When the windows were closed, the air exchange rate $k_{AER}$ was determined by continued analysis of an inert gaseous tracer ($CO_2$) injected in the room at the beginning of the experiment. For a given day, the measured $k_{AER}$ were almost constant, which allows to run the simulations with a daily average value for each experiment. Indoor temperature and humidity were measured each ten seconds. They are involved in the computation

of air viscosity, friction velocity, species diffusivities, thermal and deposition velocities, and uptake values. The durations of the experiments, average ventilation rates, minimum and maximum temperatures and humidities, and average total VOC concentration recorded are summarized in Tab. 3.

Outdoor concentrations, used as model input, are estimated by linear interpolation of outdoor measurements. VOCs were measured with a PTR-MS-ToF (Proton Transfer Reaction - Mass Spectrometer - Time of Flight) equipped with a motorized

valve rotating alternatively for 5 minutes outdoors and 10 minutes indoors, providing an outdoor VOC measurement each 15 minutes. The $O_3$ outdoor concentrations were measured at a rate of one measurement per minute. The NO and $NO_2$ outdoor concentrations were recorded on a quarter-hourly basis by the regional air-quality network AtmoSud at a station located about 1.5 km away and for which the NOx concentrations are expected to be representative of the concentrations close to the building. Outdoor HONO, HO and $HO_2$ were not measured. They are thus fixed at a constant and realistic value of 20 ppt for HONO,

$10^6$ molecules per cubic centimeter (molecules.cm$^{-3}$) for OH and $10^8$ molecules.cm$^{-3}$ for $HO_2$ (Holland et al., 2003).

Due to the PTR-ToF-MS valve rotations, an indoor VOC measurement each 15 minutes was performed with a shift of 5/10 minutes with the previous/subsequent outdoor VOC measurement. Indoor NOx were measured by chimiluminescence, HONO using a LOPAP (LOng Path Absorption Photometer), and $O_3$ by spectrophotometry. All of these instruments were placed in a separate room. The presence of instruments in the experiments room would have increased the surface available for

heterogeneous reactivity in a hardly quantifiable way, thus introducing uncertainty. $O_3$ was captured at the center of the room, at a rate of one measurement per minute. NOx were measured each second and HONO each 15 seconds. The modelled $O_3$, HONO and NOx are compared to these experimental records. The sources of these species are infiltration from outdoors and chemical reactivity, so no emission rate is considered for them.

### 3.2 Model parameters

The inorganic species measurements are considered as a benchmark to estimate the model undetermined parameters. These parameters are the building filtration factor $f$, the kinetic constants of the surface and desorption reactions (see Tab. 1), the





uptake coefficients of $O_3$, NO and HONO, the initial concentrations of the surface species $NO_{(ad)}$, $NO_{2\,(ad)}$, $HONO_{(ad)}$ and $HNO_{3(ad)}$ and, to a lesser extent, the speed of air in the room $u_{inf}$.

The speed of air in the room is assessed by measuring the homogenization time of a tracer gas injection. A styrene injection
allowed to estimate that this speed could range between 0.05 and $0.4\,\mathrm{m.s}^{-1}$. Furtaw Jr. et al. (1996) and references therein suggest the same admissible bounds for this parameter, with a value of $0.15\,\mathrm{m.s}^{-1}$ identified as reference for indoor comfort conditions (McQuiston et al., 2004). In the experiments of this paper, two fans were placed on both sides of the room, providing an active air mixing, and thus a $u_{inf}$ value likely exceeding $0.15\,\mathrm{m.s}^{-1}$.

The building filtration factor controls the pollutants fluxes from outdoor, and is completely undetermined. According to
Sarwar et al. (2002), its value ranges between 0.10 (airtight building) and 0.90 (permeable building). In the absence of measurement, it is omitted or taken as unity in most models (Sarwar et al., 2002; Carslaw, 2007; Mendez et al., 2015).

Among the surface species introduced in this model, $NO_2$ is the only one whose uptake value was determined experimentally, the other ones are adjusted by the user. The value of $\gamma_{NO}$ is expected to be low, given that all the models consider a zero deposition velocity for NO, following Nazaroff and Cass (1986). The uptake coefficients provided for NO, HONO and $O_3$ are
supposed to be the uptake values at $T_{ref} = 296$ K, $H_{ref} = 40\%$, $I_{ref} = 8.5$ W/m$^{-2}$, at a concentration of 40 ppb. Their variations are parameterized with the same relationships as the ones obtained for $NO_2$ (see section 2.4.3).

According to Ramazan et al. (2004), water competes with $HONO_{(ad)}$ for surface sites, and displaces $HONO_{(ad)}$ toward gas phase as the surface water vapor increases. The higher the water vapor, the more $HONO_{(ad)}$ desorbs. On the opposite, the lower the water vapor, the more $HONO_{(ad)}$ is available to react with other sorbed species such as NO. In this study, it is assumed
that the same holds for the other adsorbed compounds. As no information on the surface water concentration is available, the desorption reactions are parameterized as a function of the room humidity. The desorption kinetic constants are defined as

$$k_{i,(ad)} = \alpha_i k_{H,i} n_{H_2O} \tag{34}$$

where $n_{H_2O}$ is the number of water molecules in the room computed from the water mass fraction (absolute humidity), $k_{H,i}$ is the Henry's Law constant of compound $i$ [bar.mol.kg$^{-1}$] and $\alpha_i$ is a tunable variable [kg.bar$^{-1}$.mol$^{-1}$.s$^{-1}$.molecules$^{-1}$]. The
value of $k_{H,i}$ characterizes the compound affinity for water. The temperature range of these experiments (see Tab. 3) is considered as sufficiently narrow to neglect the dependence of $k_{H,i}$ on temperature. At $T = 298.15$ K, $k_{H,NO} = 0.0019$ bar.mol.kg$^{-1}$, $k_{H,NO_2} = 0.012$ bar.mol.kg$^{-1}$ and $k_{H,HONO} = 49$ bar.mol.kg$^{-1}$ (Linstrom and Mallard). For simplicity, in the rest of this paper, we will denote $k'_{i,(ad)} = \alpha_i k_{H,i}$.

Another element that can be considered as uncertain is the stoichiometry of the $NO_2$ hydrolysis reaction. Generally, it is
assumed that $HONO_{(ad)}$ and $HNO_{3(ad)}$ are formed with equal yields, but to date, and to the authors' knowledge, no experimental validation of this production ratio is available. By denoting $NO_{2\,(ad)} \rightarrow \beta_{HNO_3} HNO_{3(ad)} + \beta_{HONO} HONO_{(ad)}$, variations of $\beta_{HNO_3}$ and $\beta_{HONO}$ will be considered, with the constraint that $\beta_{HNO_3} + \beta_{HONO} = 1$, to assure nitrogen conservation.



### 3.3 Initial conditions

According to Nazaroff and Cass (1986), simulations can be sensitive to changes in initial conditions over a characteristic
time which can be considered as proportional to the inverse of the air exchange rate. When the period simulated extends over
several days (Sarwar et al., 2002; Carslaw, 2007; Courtey et al., 2009), the influence of initial conditions can be neglected. In the
present study, the air renewal time ($k_{AER}^{-1}$), *i.e.* the minimum time needed to break free from the initial conditions, represents
about the half of the simulated periods, thus requiring to set the initial concentrations carefully. The RACM2 organic and
inorganic compounds concentrations are initialized using the concentrations measured at the start time of the experiments,
summarized in Tab. 4. However, this is not sufficient to initiate the radicals chemistry which is influenced by a variety of
species, including VOCs that were not measured because unidentifiable, or under the detection limit of the PTR-MS-ToF.
Without proper initialization, the chemistry of radicals is absent from the start of the experiment, which damages the inorganic
chemistry and thus the comparison between model and experiments.

To assess the initial concentrations of the species that were not detected, a simulation is run while forcing the organic and
inorganic compounds to follow their measured values, for a duration $d_{init}$. Then, a new simulation is launched by assigning the
concentrations obtained at the end of $d_{init}$ to the compounds that were not measured; the other ones are again initiated using the
concentrations measured at the start time of the experiments, reported in Tab. 4. With these new initial conditions, the simulated
concentrations of radicals are higher than in the initial simulation, as shown by the variations of the NOx profiles (see Figs. A1-
A3 in Appendix), which are strongly influenced by the concentrations of radical species. This procedure is repeated iteratively.
The correspondence between these simulation runs and the NOx concentrations is assessed by computing the Mean Normalized
Gross Error (MNGE) over the 5000 first seconds of the simulation run. The time $d_{init}$ is not the same for all experiments, but
is fixed for a given experiment. This time is chosen depending on the speed of convergence of the simulation runs, which
increases with increasing $k_{AER}$ and decreases with increasing VOC concentration (see Tab.3). This duration amounts to 2300 s
for Experiment 1, 1900 s for Experiment 2 and 4200 s for Experiment 3. Proper initial concentrations are considered as achieved
when the MNGE with respect to NO and NO$_2$ stabilizes or reaches a minimum.

In Fig. 5, the simulations performed by initializing only the compounds that were quantified experimentally are labelled as
"without radical initialization". The simulations performed by initializing all the species, following the procedure explained
above, are labelled as "full initialization". The "full initialization" completely modifies the NO and HONO profiles, as well as
the first part of the NO$_2$ profile.

It is clear that all the compounds do not contribute equally to the radical chemistry. Namely, the initialization of compound
PPN (PeroxylPropionyl Nitrate) allows to bridge about half of the gap between the simulations "without radical initialization"
and the simulations with "full initialization". Among the 68 RACM2 compounds that were not detected during the experiments
but for which the model predicts a non-zero concentration, only 6 need to be initialized to get proper radical chemistry. These
are PAN (Peroxyacetyl nitrate and higher saturated PANs), MGLY (Methylglyoxal and other alpha-carbonyl aldehydes) DCB1,
DCB2 and DCB3 (Unsaturated dicarbonyls), and PPN. The initialization of these compounds in addition the species measured
experimentally provides the same effect on the inorganic compounds as the "full initialization".





It can be inferred from this section that a careful assessment of the initial concentrations, including for compounds that were not experimentally detected, is mandatory for such type of study.

## 4    Reference simulations

The parameters presented in section 3.2 are calibrated to reach the best correspondence between experimental data and simulations as possible. The filtration factor $f$ varies between experiments depending on wind conditions. The speed of air and the stoichiometry of the $NO_2$ heterogeneous hydrolysis do not vary between experiments. However, the values of the surface kinetic reaction rates, desorption rates and uptake coefficients may vary between experiments, because of differences in wall covering. Therefore, the parameters are first adjusted for each experiment independently. This set of optimized parameters is

denoted "Set 1". To determine parameter values usable in a wide range of conditions, the parameters are then varied to use the same values of surface kinetic reaction rates $k_S$ for all experiments, but letting the desorption rates $k'_{i,\,(ad)}$ vary with experiment. This leads to the set of parameters "Set 2". Finally, the set of parameters "Set 3" corresponds to parameters adjusted to use the same values for all experiments. Note that in the first experiment, the desorption constant $k'_{NO\,(ad)}$ still requires a lower value than the common one. In the rest of this paper, "reference simulations" will denote the simulations obtained with the "Set 3"

parameter values, while "optimized simulations" will denote the simulations obtained with the "Set 1" parameter values. All the parameter values are listed in Tab. 5.

Figs. 6, 7 and 8 present the three sets of simulations for the three experiments. In these graphs, the solid lines denote the concentrations simulated in the sunlit box, the dashed lines the concentrations in the shaded box. These two curves are identical. The NO, $NO_2$ and NOx outlying dots observed at around 10h20 and 12h10 for Experiment 1, and 12h45 and 14h20 for

Experiment 3, are sporadic outdoor measurements, and are thus not simulated by the model. During Experiment 1, an artificial $NO_2$ injection of about 56 ppb (including a few ppb of NO) was performed at 13h30. The simulated NO and HONO outbreaks generated by this $NO_2$ injection exceed the concentrations measured experimentally; they arise from surface chemistry and cannot be cancelled out by changing the parameters without damaging the simulated profiles before the injection.

The similarity between simulations and experiments is quantified by computing, for the four modelled inorganic compounds,

the relative error between the average measured and simulated concentrations, the Root-Mean-Square Error (RMSE), the Mean Normalized Gross Error (MNGE) and the Mean Normalized Bias Error (MNBE), as presented in Tab. 6. For the first experiment, these indicators are computed over the period preceeding the $NO_2$ injection only.

The NOx concentration is quite well modelled in Experiments 2 and 3, with a MNGE of 4-6%. For $NO_2$, the MNGE is about 22% in Experiment 2 while it reaches about 28% in Experiment 3. Regarding NO, the MNGE is about 17% in Experiment

2, and 35% in Experiment 3. In Experiment 1, the NOx concentrations are systematically underestimated, with an MNBE of -62% in the first part of the simulation. In the second part, the $NO_2$ decay following the $NO_2$ injection is quite well replicated using the optimized parameters. For all experiments and all sets of parameters, $O_3$ is underestimated with a relative error ranging between 50% and 60%. HONO exhibits excellent statistics for Set 1 and Set 2, with an MNGE of 1% to 9% in the





three experiments. Yet, using the common parameters (Set 3), HONO is underestimated in Experiment 2 (-8% MNBE), and strongly overestimated in Experiments 1 and 3 (94% and 78% MNBE respectively).

By comparing the simulations by sets of parameters, we can observe that for Experiments 2 and 3, the Set 1 and Set 2 parameters lead to very similar concentrations, whereas the use of common values for desorption constants (Set 3) increases the error on HONO. The Set 1 results stands as the best correspondence that can be achieved. In these experiments, the discrepancy between model and measurements observed using common parameters (Set 3) can be cancelled out by varying the NO and HONO desorption constants (Set 2). In the first experiment, the NOx curves are identical for Set 2 and Set 3. They differ from the Set 1 curves because in that case, the parameters were optimized to replicate the $NO_2$ decay and mitigate the NO release following the injection.

It can be concluded from this section that by combining homogeneous and heterogenous reactivity, the $H^2I$ model is able to replicate the inorganic chemistry. The model parameters which could not be determined experimentally were treated as tunable parameters. The model yields satisfying results using the same parameter values for all experiments (Set 3), apart from the NO profile in Expriment 1, where a lower $k'_{NO\,(ad)}$ is needed. The model results can be improved up to the best achievable results (Set 1), by merely varying the NO and HONO desorption rates (Set 2).

## 5 Sensitivity study

The purpose of this section is to investigate the relative influence of the model parameters. This section focuses on how the inorganic chemistry is influenced by surface reactions. As Experiment 1 is a particular case ($NO_2$ injection), only Experiments 2 and 3 are considered for these tests. When a parameter is varied, the simulated results are presented for one experiment only, as the conclusions are identical for both experiments. Each tunable parameter is investigated independently. The parameters which are not varied are given the same values as the optimized parameters (Set 1) listed in Tab. 5.

The initial concentrations of the gas-phase species are determined using the procedure described in section 3.3 and are summarized in Tab. 4. The sorbed species $NO_{(ad)}$, $NO_{2(ad)}$, $HONO_{(ad)}$ and $HNO_{3(ad)}$ do not undergo the same processes as the gas-phase species. Their evolution is driven by chemical reactivity only. Since box exchange is disabled for these species, these species profiles in the shaded and sunlit volumes are well distinct, as shown by all the figures presented in this section. At the start of the simulations, the concentrations of these species rapidly converge to the values determined by surface chemistry. Proper sorbed species initial concentrations are thus easy to set, after running a couple of simulations. For a given experiment, the initial concentrations remain unchanged whatever the parameter investigated.

### 5.1 Filtration factor and speed of air

Indoor chemistry is influenced by the filtration factor and the speed of air, which are parameters characteristic of the environment. The filtration factor controls the input flux of outdoor pollutants, while the speed of air governs deposition. This section aims at assessing to what extent these parameters affect the overall inorganic concentrations.





### 5.1.1 Speed of air

The gas-phase and adsorbed inorganic compounds simulated with a speed of air $u_{\mathrm{inf}}$ ranging from $0.06\,\mathrm{m.s}^{-1}$ to $0.30\,\mathrm{m.s}^{-1}$ are presented in Fig. 9. This range of variations corresponds to the range of expected values described in section 3.2. The modelled $NO_2$ and $O_3$ concentrations decrease with increasing $u_{\mathrm{inf}}$ while the modelled NO and HONO concentrations increase with increasing $u_{\mathrm{inf}}$. These opposite behaviours derive from the type of the source and processes contributing to these species 510 concentrations at most. It can be easily inferred from Eqs. (16-20) that the larger $u_{\mathrm{inf}}$, the larger the deposition on surfaces. When $u_{\mathrm{inf}}$ is increased, the $O_3$ surface removal increases. As the main source of $O_3$ is transport from outdoor (Weschler and Shields, 1996), this loss of $O_3$ is not counterbalanced by another source, which results in a decrease of $O_3$ with increasing $u_{\mathrm{inf}}$. A contrario, HONO is mainly produced by heterogeneous processes which are predominant indoors. The increase of $u_{\mathrm{inf}}$ enhances the $NO_2$ deposition, and thus the HONO production by the $NO_2$ hydrolysis on surfaces. Indoor NOx concentrations are 515 influenced by both outdoor concentrations and surface chemistry (Weschler et al., 1994). In these experiments, variations with $u_{\mathrm{inf}}$ indicate that the main $NO_2$ source is outdoor infiltration whereas NO is mainly produced by heterogeneous processes. The value retained for $u_{\mathrm{inf}}$ is $0.24\,\mathrm{m.s}^{-1}$, considering it is large enough to achieve an effective deposition and stimulate secondary chemistry, while fulfilling the criterions presented in section 3.2. As this value is the result of controlled air mixing by fans, it is kept unchanged from one experiment to the other.

### 5.1.2 Filtration factor

Contrary to $u_{\mathrm{inf}}$, increasing $f$ leads to an increase of $NO_2$ and $O_3$, but also of NO and HONO (see Fig. 10). It must be underlined that increasing $f$ increases the intake of outdoor pollutants, but not the air exchange rate which remains unchanged. Increasing the fraction of air exchange with outdoor increases the concentration of outdoor pollutants like $O_3$ and $NO_2$. In turn, the increased $NO_2$ concentration fosters the secondary production of HONO and NO. For these experiments, an average value of 525 0.30 appears appropriate to match the overall amount of NOx, and by extension the amount of HONO. As mentioned in section 4, differences between experiments can be caused by variations in outdoor wind conditions.

### 5.2 $NO_2$ heterogeneous hydrolysis : $NO_{2\,(ad)} \rightarrow 0.5\ HNO_{3(ad)} + 0.5\ HONO_{(ad)}$

The influence of the $NO_2$ heterogeneous hydrolysis is now investigated, by varying its stoichiometry and kinetic rate.

### 5.2.1 Stoichiometry

As introduced in section 3.2, the stoichiometry of the $NO_2$ hydrolysis reaction is uncertain. Fig. 11 presents the evolution of the inorganic concentrations with different yields $\beta_{HNO_3}$ and $\beta_{HONO}$. Because HONO concentrations are underestimated when $\beta_{HNO_3} = \beta_{HONO} = 0.5$, the ratio $\beta_{HNO_3}/\beta_{HONO}$ is kept $< 1$ so that $HONO_{(ad)}$ is always more produced than $HNO_{3(ad)}$. When $\beta_{HNO_3}/\beta_{HONO}$ tends to one, the production of $HONO_{(ad)}$ and $HNO_{3(ad)}$ by the $NO_2$ hydrolysis gets balanced, which favours the NO production by reaction S2. As NO increases, $NO_2$ increases by equilibration through homogeneous chemistry. When the 535 ratio $\beta_{HNO_3}/\beta_{HONO}$ tends to one, less $HONO_{(ad)}$ is available for desorption, and less gas-phase HONO is released. Conversely,





when the ratio $\beta_{HNO_3}/\beta_{HONO}$ is decreased, the enhanced $HONO_{(ad)}$ production fosters the HONO desorption, leading to higher HONO gas-phase concentration. It is noteworthy that very small variations of $\beta_{HNO_3}/\beta_{HONO}$ generates significant variations in NOx, and particularly HONO. Note that $O_3$ is mainly controlled by transport from outdoor, and is thus not affected by these parameters. The values $\beta_{HNO_3} = 0.47$ and $\beta_{HONO} = 0.53$ allow to match a consistent HONO production without differing too

much from the classical stoichiometry assumed for this reaction. They are kept unchanged from one experiment to the other.

### 5.2.2   Surface NO$_2$ conversion

$NO_2$ is adsorbed on surfaces at a rate which is determined by the transport velocity toward surfaces, and by the $NO_2$ uptake of surfaces. Once $NO_2$ is adsorbed, it is converted, in the presence of water, to form HONO and $HNO_3$, at a kinetic rate $k_{S1}$ which is highly uncertain. The larger $k_{S1}$, the more rapid the conversion and the larger the HONO production. The same holds

for NO which is produced by secondary reaction of $HONO_{(ad)}$ with $HNO_{3(ad)}$. In turn, the NO increase enhances $NO_2$ by homogeneous reactivity, thus providing new $NO_2$ available for adsorption. However, Fig. 12 shows that HONO concentrations do not vary much when $k_{S1}$ is increased above a threshold value of $0.003\,\mathrm{s}^{-1}$. As $k_{S1}$ is increased above this value, the $NO_{2(ad)}$ concentration tends to zero. When $k_{S1}$ is decreased, the $NO_{2(ad)}$ hydrolysis is slowed down, which decreases the $HONO_{(ad)}$ reservoir and thus curb the NOy (NOx + HONO) heterogeneous production. The sensitivity of this parameter is large. The

threshold value $k_{S1} = 0.003\,\mathrm{s}^{-1}$ is retained, as it maximizes the $NO_{2(ad)}$ conversion.

### 5.3   NO secondary formation

According to section 5.1.1, NO is mainly produced by secondary chemistry in these experiments. In this section, the importance of two reactions forming $NO_{(ad)}$ is studied.

### 5.3.1   $HONO_{(ad)} + HNO_{3(ad)} \rightarrow 2\,NO_{(ad)}$

First, $NO_{(ad)}$ can be produced by the reaction of $HONO_{(ad)}$ with $HNO_{3(ad)}$, at a kinetic rate $k_{S2}$. Fig. 13 shows that increasing $k_{S2}$ enhances the formation of $NO_{(ad)}$, and thus its release to the gas-phase. As less $HONO_{(ad)}$ is available, the gas-phase HONO concentration is lowered. A contrario, if $k_{S2}$ is lowered, the reaction of $HONO_{(ad)}$ with $HNO_{3(ad)}$ gets slow compared to the desorption of $HONO_{(ad)}$, and most of the $HONO_{(ad)}$ is released in the gas phase. In turn, the $NO_{(ad)}$ production gets too low to maintain a NO release allowing to reach the measured concentrations. These results indicate that there is a competition

between the desorption of $HONO_{(ad)}$ and the reaction of $HONO_{(ad)}$ with $HNO_{3(ad)}$ to consume $HONO_{(ad)}$. When calibrating $k_{S2}$ and $k'_{HONO\,(ad)}$, a balance between these two reactions must be found, to obtain consistent concentration profiles for both HONO and NO. Similarly to $k_{S1}$, the parameter $k_{S2}$ seems to be a very sensitive one, as it significantly affects NO, HONO, and also $NO_2$ through the means of NO. The value $k_{S2} = 10^{-13}\,\mathrm{s}^{-1}$ retained for the reference simulations is a compromise between the optimized values calibrated for each experiment.





### 5.3.2 HONO$_{(ad)}$ → 0.5 NO$_{(ad)}$ + 0.5 NO$_{2(ad)}$

Another NO$_{(ad)}$ formation pathway is the autoionization of HONO$_{(ad)}$. The larger the kinetic constant $k_{S3}$ of this reaction is, the larger the HONO$_{(ad)}$ conversion into NO$_{(ad)}$ and NO$_{2(ad)}$ is, and the lower the HONO$_{(ad)}$ reservoir available for desorption is. However, contrary to reaction S$_2$, the autoionization of HONO$_{(ad)}$ does not compete with the desorption of HONO$_{(ad)}$. It can be observed in Fig. 14 that when $k_{S3} < 10^{-5}$, the effect of this reaction vanishes, meaning that the HONO concentrations are only determined by $k_{S2}$ and $k'_{\text{HONO}(ad)}$ : as discussed in the previous subsection, decreasing $k_{S2}$ with $k_{S3} = 10^{-5}$ enhances the release of HONO and cuts off the production of NO$_{(ad)}$, showing that reaction S$_2$ and desorption weigh in on the depletion of HONO$_{(ad)}$ at equal level. If $k_{S3}$ is raised above that value, the gas-phase NOx concentrations increase, but it gets more difficult to lift HONO up to the concentrations measured experimentally, indicating that $k_{S3}$ should not be increased too much when calibrating the model kinetic constants. The threshold value $k_{S3} = 10^{-5}\,\text{s}^{-1}$ is kept for this work.

### 5.4 NO$_2$ regeneration : NO$_{(ad)}$ + HNO$_{3(ad)}$ → NO$_{2\,(ad)}$ + HONO$_{(ad)}$

As mentioned in the introduction, NO$_{2(ad)}$ and HONO$_{(ad)}$ can be regenerated through the reaction of NO$_{(ad)}$ with HNO$_{3(ad)}$. Fig. 15 shows that the larger the kinetic constant $k_{S4}$ of this reaction is, the lower the NO and NO$_2$ concentrations are, but the larger the HONO concentration is. Increasing $k_{S4}$ promotes the consumption of NO$_{(ad)}$, thus curbing the release of NO to the gas phase. The production of HONO$_{(ad)}$ is enhanced, which in turn stimulates the HONO release. The production of NO$_{2(ad)}$ is also enhanced by reaction S$_4$, which should increase the NO$_{2(ad)}$ reservoir. However, reaction S$_1$ competes with desorption in the depletion of NO$_{2(ad)}$. As reaction S$_1$ reduces the surplus of NO$_{2(ad)}$ produced by reaction S$_4$, there is no increase of the NO$_{2(ad)}$ reservoir. The release of NO$_2$ is not enhanced, and the gas-phase NO$_2$ concentration equilibrates with the decreased NO concentration. In turn, increasing $k_{S4}$ lowers the NO$_2$ concentration, whereas it raises the HONO one.

When $k_{S4}$ is decreased below $10^{-15}$, the effect of this reaction on the inorganic concentrations vanishes, showing that like reaction S$_3$, reaction S$_4$ does not compete with another reaction, contrary to reactions S$_1$ and S$_2$. Increasing $k_{S4}$ increases the HONO concentration, but lessens the NOx level at the same time, which is unfavourable beyond a certain threshold. This suggests that the kinetic constant $k_{S4}$ should remain low enough to keep reaction S$_4$ upstage, like reaction S$_3$. In these simulations, the value $k_{S4} = 2 \times 10^{-15}$ allows to back the HONO production while meeting this requirement.

### 5.5 Desorption constants and uptake values

The desorption constants and uptake coefficients drive the exchanges between the adsorbed and the gas phases. They are now examined.

#### 5.5.1 Nitrogen dioxide

The uptake coefficient $\gamma_{\text{NO}_2}$ is the only parameter characteristic of the heterogeneous chemistry of inorganic compounds that was determined experimentally for the paint boards. This parameter is not modified in Experiments 2 and 3 where the paint boards were used. Here, the sorption dynamics of NO$_2$ is only modified through the desorption constant $k'_{\text{NO}_2\,(ad)}$. It can be





inferred from Fig. 16 that the $NO_2$ concentration is not very sensitive to $k'_{NO_2\,(ad)}$, as this latter must be varied over several orders of magnitudes to observe significant changes in $NO_2$ concentrations, likely because the main source of $NO_2$ is transport from outdoor, and not secondary chemistry. Increasing $k'_{NO_2\,(ad)}$ decreases the $NO_{2(ad)}$ reservoir. As less $NO_{2(ad)}$ is available, less $HONO_{(ad)}$ is produced by reaction $S_1$, thus resulting in a decreased release of HONO to the the gas phase. To maintain

sufficient $NO_2$ adsorption and large enough HONO concentration, the parameter $k'_{NO_2\,(ad)}$ should be kept low. It is set to $k'_{NO_2\,(ad)} = 10^{-22}\,\text{s}^{-1}.\text{mlc}^{-1}$ in the reference simulations.

### 5.5.2   Nitric oxide

To date, all the box models (Sarwar et al., 2002; Carslaw, 2007; Courtey et al., 2009; Mendez et al., 2015) assume a zero deposition velocity for NO after the values reported by Nazaroff and Cass (1986). A zero deposition velocity corresponds to

an uptake coefficient close to zero, thus preventing the molecules from colliding with surfaces and getting adsorbed.

    Fig. 17 investigates the sensitivity of inorganic concentrations to $\gamma_{NO}$, which is varied between $\gamma_{NO} = 10^{-8}$ and the maximum value $\gamma_{NO} = \infty$. The coefficient $\gamma_{NO} = \infty$ is obtained by assuming that all the collisions are efficient, that is $k^j_{DEP,i} = k^j_{tran,i}$. When $\gamma_{NO} = \infty$, the NO deposition is only limited by transport to the surface. Apart from the beginning of the simulations, no significant variation in NO concentration is observed between the extreme values investigated, showing that in this experiment,

deposition has a negligible contribution to the gas-phase NO concentration. The first part of the simulated profile can be improved by about 10% using the lowest $\gamma_{NO}$ value.

    The concentration variations with desorption constant $k'_{NO\,(ad)}$ are presented in Fig. 18. When $k'_{NO\,(ad)}$ is increased, $NO_{(ad)}$ evaporates toward gas phase. A contrario, when $k'_{NO\,(ad)}$ is decreased, the release of NO is less efficient and the NO concentration decreases, leading to a decrease in $NO_2$ concentration. Then, since less $NO_2$ is available on surfaces for hydrolysis, less HONO

is produced. As a result, decreasing $k'_{NO\,(ad)}$ too much levels down the three NOy compounds concentrations in the gas phase.

    In turn, the parameters $\gamma_{NO}$ and $k'_{NO\,(ad)}$ should be fixed so as to keep NO in gas-phase form preferentially, which corroborates the use of a very low deposition velocity, as done in the literature. In this paper, an uptake coefficient $\gamma_{NO} = 8 \times 10^{-9}$ is chosen for all the simulations. The desorption constant $k'_{NO\,(ad)}$ is set to $8 \times 10^{-21}\,\text{s}^{-1}.\text{mlc}^{-1}$ for Experiments 2 and 3, while a lower value $2 \times 10^{-22}\,\text{s}^{-1}.\text{mlc}^{-1}$ appears required to simulate Experiment 1.

### 620   5.5.3   Nitrous acid

The adsorption/desorption dynamics of HONO is investigated by varying the uptake coefficient $\gamma_{HONO}$ (Fig. 19) and the desorption constant $k'_{HONO\,(ad)}$ (Fig. 20). Contrary to NO, small variations of uptake coefficient and desorption constant make the HONO concentration vary a lot. HONO is not brought by transport from outdoor and is only produced by secondary chemistry, which justifies the critical importance of these parameters controlling the transfers between the homogeneous gas-phase and

surfaces. Increasing $\gamma_{HONO}$ leads to increase the $HONO_{(ad)}$ reservoir and therefore to decrease the gas-phase HONO. When $\gamma_{HONO}$ tends to zero, the gas-phase HONO concentration is determined by the desorption constant $k'_{HONO\,(ad)}$ only. As the HONO concentration is sensitive to both $\gamma_{HONO}$ and $k'_{HONO\,(ad)}$, a balance between these parameters must be found. An increase in $\gamma_{HONO}$ can compensate an increase in $k'_{HONO\,(ad)}$, and reciprocally, a decrease in $\gamma_{HONO}$ must be associated with a decrease





in $k'_{\mathrm{HONO\,(ad)}}$. Several choices (large $\gamma_{\mathrm{HONO}}$ and $k'_{\mathrm{HONO\,(ad)}}$ *vs.* low $\gamma_{\mathrm{HONO}}$ and $k'_{\mathrm{HONO\,(ad)}}$) allow to simulate the HONO concen-

tration correctly. However, the time variations of the HONO concentration may behave differently depending on the chosen

set of parameters. In this example, if both $\gamma_{\mathrm{HONO}}$ and $k'_{\mathrm{HONO\,(ad)}}$ are large, the HONO concentration tends to bend at the end

of the simulation, whereas a monotonous increase is observed in the experiment. This suggests that low values of $\gamma_{\mathrm{HONO}}$ and

$k'_{\mathrm{HONO\,(ad)}}$ are better suited. The values $\gamma_{\mathrm{HONO}} = 2 \times 10^{-8}$ and $k'_{\mathrm{HONO\,(ad)}} = 5 \times 10^{-22}\,\mathrm{s^{-1}.mlc^{-1}}$ are retained for the reference

simulations.

Finally, while significant variations of HONO concentrations are observed, changes in NO and $NO_2$ are imperceptible, thus

showing the poor correlation between HONO and NOx concentrations in these experiments with anti-UVs windows.

### 5.5.4    Ozone

In all the simulations presented above, the $O_3$ concentrations are not altered by any change in NOx and HONO concentrations.

This is an expected behaviour, considering that the main source of $O_3$ is transport from outdoor, and that its main sink is

deposition. Fig. 21 shows that variations in uptake coefficient $\gamma_{O_3}$ modify the $O_3$ concentration significantly, and also the NO

concentration by means of homogeneous chemistry. When $\gamma_{O_3}$ tends to zero, the $O_3$ concentration increases up to a value very

close to the experimental one, whereas the NO concentration decreases to concentrations much lower than those observed.

With an uptake coefficient $\gamma_{O_3} = 10^{-6}$ and no desorption reaction for $O_3$, both O3 and NO are correctly modelled : albeit

lower than the experimental record, $O_3$ remains within 60% of the measured concentration, while the NO RMSE keeps close

to one.

### 6    Modelling high $NO_2$ concentrations : focus on Experiment 1

When analyzing the optimized simulations, it can be noticed that the parameters fitting Experiments 2 and 3 are rather similar,

but can significantly differ from some of those fitting Experiment 1, especially the ones controlling the adsorption/desorption

of the NOy compounds. The difficulty in simulating Experiment 1 (Fig. 6) lies in handling the fast transition from a moderate

$NO_2$ concentration to a very high one. To prevent the HONO and NO secondary productions from rocketing after the $NO_2$

injection, $k'_{\mathrm{NO\,(ad)}}$ was decreased and $\gamma_{\mathrm{HONO}}$ was increased so as to limit the NO and HONO releases. To preserve satisfying

NO and HONO levels before the injection, the initial concentrations of $NO_{(ad)}$ and $HONO_{(ad)}$ were pushed up in order to

counterbalance the weaker release of these species.

In spite of the above, changing the model parameters did not allow to completely mitigate the NO and HONO breakouts

caused by the massive $NO_2$ supply. Previously, the modelling of the HONO production in high $NO_2$ conditions, *i.e.* $NO_2$ con-

centrations exceeding 25 ppb, was already pointed out by Mendez et al. (2017) as a challenging issue. Like in this study, the

experimental HONO step up caused by the $NO_2$ injection was moderate, which existing model failed to replicate, by overesti-

mating the HONO increase. Mendez et al. (2017) proposed to cope with that by introducing a compound SURF representing

the number of sites available for $NO_2$, thus limiting the amount of $NO_{2(ad)}$ for surface hydrolysis.





In this paper, a similar solution is implemented by extending the definition of SURF to all the surface compounds introduced in this model. Here SURF represents the number of sites available for NO, HONO, $NO_2$ and $HNO_3$. SURF is incorporated in the adsorption/desorption reactions, but only modifies the kinetics of these reactions when its "concentration" is less than unity. In other words, as long as lots of surface sites are available, the sorption dynamics behaves as usual, but as soon as the surface approaches saturation, the kinetics of the adsorption reactions is increased of one order, in addition to be slowed down 665 by the SURF "concentration" less than unity.

   The resulting profiles are presented in Fig. 22, using the same parameters as the optimized simulation. The main differences with Mendez et al. (2017) are that the NO, $NO_2$ and $O_3$ concentrations are not fixed to their measured values, and that the first part of the HONO profile matches the experimental data. In this test, the $NO_2$ injection makes the three NOy compounds uprise. The magnitude of this uprise is determined by the initial value of the SURF "concentration". When this value is very 670 large, the concentration profiles converge to the optimized simulation (Fig. 6). It appears that limiting the sorption of $NO_2$ and related species generates an excess of gas-phase NOy which largely exceeds the decrease of NOy produced by heterogeneous chemistry. If the overestimation of the NO and HONO secondary formation cannot be mitigated by limiting the adsorption of gas-phase species, the most probable way to concile models and experiments may be to search for more complexe surface processes that could account for that.

**7 Discussion**

Considering the number of parameters involved in this model, a given simulation result is likely reachable by several parameters combinations. Additionally, some parameters allowing to replicate the records presented in this study are typical of the room of the experiments and would be hardly transferable to other indoor environments. These are the filtration factor, the speed of the air, the desorption constants and the uptake coefficients. Although these parameters are basically environment-dependent 680 and non-unique, multiple conclusions can be drawn from these tests regarding the general principles of IAQ modelling.

   The air mixing velocity introduced like in Mendez et al. (2017) appears as a critical parameter for the four main species leading the inorganic chemistry. The building filtration factor, generally taken as unity because barely studied, also stands as a determining factor. This parameter should not be confused with the ventilation rate ($k_{AER}$) which encompasses both the leaks to outside and to the rest of the building. The building filtration factor is the fraction of air influx coming from outside the 685 building. In these experiments, its value is far from unity, thus featuring airtight windows. Using a filtration factor of unity would have lead to overestimate the intake of outdoor pollutants, thus mitigating the importance of heterogenous phenomena.

   The surface hydrolysis of $NO_2$ produces sorbed HONO and $HNO_3$ which further react to produce sorbed NO. The kinetics of this reaction is determined by its kinetic rate $k_{S1}$, and also by the $NO_2$ adsorption/desorption reactions which control the variations of the $NO_{2(ad)}$ reservoir. The $NO_{2(ad)}$ concentration variations largely influence the variations of the other sorbed 690 species concentrations through the means of reaction $S_1$, which could be considered as the cornerstone of indoor heterogeneous chemistry. Interestingly, a very small adjustment in the stoichiometry of reaction $S_1$ allows to increase the HONO secondary production significantly in all experiments.





The NO secondary production mainly derives from reaction $S_2$ which competes with the HONO desorption. The kinetic rates of these reactions can be influenced by the nature of the surface materials. The release kinetics of these species flow from
their sorbed concentrations, which likely depend on a variety of parameters. Such environment-dependence could explain why the yields in HONO and NO reported in the literature can vary a lot from one study to another (Finlayson-Pitts et al., 2003). Reactions $S_3$ and $S_4$ can influence the NOy concentration time variations, but their impacts seem less predominant.

Regarding the sorption dynamics, as almost entirely produced by heterogeneous chemistry, HONO is extremely sensitive to the values of its uptake coefficient and desorption constant. Obviously, these parameters have antagonist effects which can
neutralize each other. The proper calibration of these parameters can be oriented by the shape of the sorbed species profile, with respect to the gas-phase experimental one. Contrary to HONO, the NO concentration does not seem affected by the uptake coefficient $\gamma_{NO}$. Similarly to the radical HO, NO is likely too reactive to be affected by deposition. Therefore, like done in the literature, a very small deposition velocity appears well suited for NO, whatever its uptake coefficient.

The simulations presented in section 4 show that the desorption constants are the parameters which are the most difficult
to set, especially for NO and HONO. The problem is less striking for $NO_2$ since the main source of $NO_2$ is transport from outdoor in these experiments. By using a common value of $k'_{HONO\,(ad)}$ for the three experiments (reference simulations), the simulated HONO concentrations show an overestimation of 95% and 78% for Experiments 1 and 3. Using the same $k'_{NO\,(ad)}$ in Experiments 2 and 3, the simulated NO concentrations remain as similar to the measured ones as using the optimized parameters. However, it is impossible to use the same value in Experiment 1 without making the NO concentration increase
well above the measured values.

Small variations in uptake coefficients and desorption constants can be supported by differences in wall cladding between experiments. In the first one, walls were naked, whereas in the subsequent ones, walls were covered by boards freshly coated with a paint made of the same organic matrix. According to Finlayson-Pitts et al. (2003), the composition of the surface film of water can play an important role in determining the yields of NO and HONO, but the nature of the underlying material
should not influence this chemistry, unless it is sufficiently reactive to modify the composition of the surface film. The surface topology can also influence the material adsorption capacity : experiments conducted by Wainman et al. (2001) to study the influence of the surface nature on the $NO_2$ hydrolysis showed that HONO concentrations were significantly enhanced when synthetic carpet was used instead of Teflon surfaces. They suggested this was caused by the greater surface quantity provided by the carpet fibers, allowing more room for the reaction to occur. In this study, it could be argued that differences in roughness
between the walls and the paint boards, combined with differences in uptake values, may account for variations of surface sorption capacity between the first and next experiments. However, these elements are not sufficient to support a difference in $k'_{NO\,(ad)}$ of almost three orders of magnitude (see Set 1 in Tab. 5).

In light of this, it can be inferred that using a more sophisticated parameterization of the desorption reactions may be a possible way to improve this model. Namely, it could be necessary to take into account the multi-layer organization of the
surface film, or the migration processes from the surface materials to the interface between the surface film and the gas phase. Such improvement may also be a solution to the problems observed after the $NO_2$ injection during Experiment 1. Like Mendez et al. (2017), we observe that the measured increase of HONO after the $NO_2$ injection is moderate, and like the models tested by





Mendez et al. (2017), the H$^2$I model overestimates this increase of HONO concentration. Implementing a deposition saturation effect did not allow to improve the model's performance, but it can be hypothesized that a release limitation owing to the surface film structure may do so.

Finally, apart from the particular case of the HONO secondary production in high NO$_2$ conditions, the H$^2$I model successfully replicates the chemistry of inorganic species.

## 8    Conclusions

In this paper, a new numerical model combining homogeneous and heterogeneous chemistry is implemented and proved able to replicate the concentrations of inorganic compounds. For the first time, O$_3$, HONO and NOx species are simulated all at once and compared to experimental records acquired in a real room. The specificity of this model is to incorporate secondary reactions which were highlighted by laboratory studies but which are still absent from numerical models. It is also the first two-box model allowing to consider the variations of direct and indirect light throughout the day. The comparison between the simulation results and experimental data allowed to tune the model parameters, which lead to several conclusions : (i) the building filtration factor and the speed of air mixing are important parameters which should deserve more attention; (ii) for the simulation duration considered (a third of day in average), the proper assessment of initial concentrations is critical; (iii) whereas deposition and surface reactivity are treated together by current models, the distinction between sorption and surface reactions appears as essential. This distinction is based on the introduction of sorbed species which also have the possibility to desorb. To better constrain these sorption/desorption processes, there is a need of surface material characterization, especially measurements of O$_3$, NO$_2$ and HONO uptake coefficients and NO, NO$_2$ and HONO desorption constants, in various conditions of temperature, humidity and irradiation; (iv) the success of this model in simulating inorganic species largely arises from the better consideration of surface chemistry, thus highlighting its critical importance for indoor air reactivity. Whereas reactions S$_3$ and S$_4$ could be considered of secondary importance, reaction S$_2$ appears as important as reaction S$_1$ which is currently the only surface reaction taken into account by IAQ models. Reaction S$_2$ may account for unexplained variations in NO vs. HONO production ratios and should clearly be integrated. In the future, the further understanding of heterogenous phenomena will be a necessary step toward the improvement of IAQ models.

*Code availability.* Code available at https://gitlab.enpc.fr/cerea/h2i

*Author contributions.* HW and KS conceived the project. EAF, KS and HW developed the methodology. EAF and KS developed the code. EAF made formal analysis and visualization. EAF, KS and HW performed investigations. EAF wrote the original draft. KS and HW reviewed the draft. HW acquired the funding.



*Competing interests.* The authors declare that they have no conflict of interest.

*Acknowledgements.* This work was supported by the LABEX SERENADE (no. ANR-11-LABX-0064) funded by the "Investissements d'Avenir" program of the French National Research Agency. The experimental campaign from which the data shown in this paper originate was supported by the same funding. The authors would like to thank the LCE members who contributed to the acquisition of these data,
760  namely Adrien Gandolfo, Amandine Durand and Sasho Gligorovski.



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





**Table 1.** Heterogeneous reactions added to the RACM2 model. The symbol $\chi$ designates the species that undergo unimolecular decomposition. These species are the VOCs (see Tab. 2 for the list of considered VOCs), $O_3$, $NO_3$, $HNO_4$ and $H_2O_2$. Reactions $S_1$, $S_2$, $S_3$ and $S_4$ are the model equivalent of reactions R1, R3, R2 and R4.

| Reactions | Kinetic constants |
|---|---|
| *Unimolecular decomposition* | |
| $\chi \rightarrow$ | $k_\chi$ |
| *Adsorption reactions* | |
| $NO \rightarrow NO_{(ad)}$ | $k_{NO}$ |
| $NO_2 \rightarrow NO_{2(ad)}$ | $k_{NO_2}$ |
| $HONO \rightarrow HONO_{(ad)}$ | $k_{HONO}$ |
| $HNO_3 \rightarrow HNO_{3(ad)}$ | $k_{HNO_3}$ |
| *Desorption reactions* | |
| $NO_{(ad)} \rightarrow NO$ | $k_{NO(ad)}$ |
| $NO_{2(ad)} \rightarrow NO_2$ | $k_{NO_2(ad)}$ |
| $HONO_{(ad)} \rightarrow HONO$ | $k_{HONO(ad)}$ |
| *Surface reactions* | |
| $NO_{2(ad)} \rightarrow 0.5\ HNO_{3(ad)} + 0.5\ HONO_{(ad)}$ | $k_{S1}$ |
| $HONO_{(ad)} + HNO_{3(ad)} \rightarrow 2\ NO_{(ad)}$ | $k_{S2}$ |
| $HONO_{(ad)} \rightarrow 0.5\ NO_{(ad)} + 0.5\ NO_{2(ad)}$ | $k_{S3}$ |
| $NO_{(ad)} + HNO_{3(ad)} \rightarrow NO_{2(ad)} + HONO_{(ad)}$ | $k_{S4}$ |





**Table 2.** RACM2 molar mass $M$ [g.mol$^{-1}$], critical temperature $T_c$ [K] and critical molar volume $V_c$ [cm$^3$.mol$^{-1}$] of a species representing the RACM2 compound and diffusion coefficient $D$ [m$^2$.s$^{-1}$] computed following the procedure described in section 2.4.1. The references listed are available from the NIST webbook (Linstrom and Mallard).

| Compounds | $M$ | Representative species | $T_c$ | $V_c$ | Reference | D ($10^{-5}$) |
|---|---|---|---|---|---|---|
| OLT | 42 | Propene | 364.90 | 185 | Lide (2005) | 1.18 |
| OLI | 68 | Pentene | 464.80 | 298 | Lide (2005) | 0.833 |
| TOL | 92 | Toluene | 822.28 | 490 | - | 0.573 |
| XYL | 106 | m-Xylene | 617.00 | 375 | Tsonopoulos and Ambrose (1995) | 0.674 |
| HCHO | 30 | Formaldehyde | 436.48 | 100 | - | 1.61 |
| ALD | 58 | Propanal | 504.40 | 204 | Gude and Teja (1994) | 1.01 |
| KET | 86 | 2-Pentanone | 561.08 | 320 | Ambrose et al. (1974a) | 0.758 |
| ISO | 68 | Isoprene | 480.20 | 280 | - | 0.855 |
| CSL | 108 | Cresol | 695.15 | 290 | Glaser and Rüland (1957) | 0.752 |
| ORA1 | 46 | Formic acid | 588.00 | 130 | Ambrose and Ghiassee (1987) | 1.26 |
| ORA2 | 60 | Acetic acid | 590.70 | 171 | D'Souza and Teja (1987) | 1.07 |
| MACR | 70 | Methacrolein | 516.54 | 260 | - | 0.874 |
| MOH | 32 | Methanol | 512.50 | 117 | Gude and Teja (1994) | 1.45 |
| ACD | 44 | Acetaldehyde | 466.00 | 154 | Teja and Anselme (1990) | 1.22 |
| ACT | 58 | Acetone | 508.10 | 213 | Ambrose et al. (1974b) | 0.991 |
| BEN | 78 | Benzene | 562.05 | 256 | Tsonopoulos and Ambrose (1995) | 0.857 |
| PHEN | 94 | Phenol | 694.30 | 230 | Delaunois (1968) | 0.855 |
| BALD | 106 | Benzaldehyde | 695.00 | 340 | Ambrose et al. (1975) | 0.697 |
| ROH | 60 | Butanol | 563.00 | 274 | Gude and Teja (1995) | 0.862 |
| UALD | 84 | Pentenal | 548.86 | 310 | - | 0.775 |
| LIM | 136 | Limonene | 657.16 | 500 | - | 0.562 |
| $O_3$ | 48 | $O_3$ | 261.10 | 89 | Jenkins and Birdsall (1952) | 1.64 |
| NO | 30 | NO | 180.00 | 58 | Lide (2005) | 2.25 |
| $NO_2$ | 46 | $NO_2$ | 561.53 | 110 | - | 1.36 |
| HONO | 47 | $NO_2$ | 561.53 | 110 | - | 1.36 |
| $NO_3$ | 62 | $NO_3$ | 534.15 | 140 | - | 1.18 |
| $HNO_3$ | 63 | $HNO_3$ | 648.46 | 140 | - | 1.14 |
| $HNO_4$ | 79 | $HNO_4$ | 669.82 | 155 | - | 1.05 |
| HO | 17 | $H_2O$ | 647.10 | 56 | Sato et al. (1991) | 2.24 |
| $HO_2$ | 33 | $H_2O_2$ | 728.00 | 70 | Nikitin et al. (1995) | 1.69 |
| $H_2O_2$ | 34 | $H_2O_2$ | 728.00 | 70 | Nikitin et al. (1995) | 1.68 |



**Table 3.** Parameters of the experiments : total duration, $k_{AER}$, minimum and maximum temperature and humidity, type of paint, average total VOC concentration.

| Experiments | Day | Duration [h] | $k_{AER}$ | $T_{min}$ [°C] | $T_{max}$ [°C] | $H_{min}$ [%] | $H_{max}$ [%] | Type of paint | VOC [ppbC] |
|---|---|---|---|---|---|---|---|---|---|
| Experiment 1 | 27 October | 8.7 | 0.25 | 22.7 | 26.7 | 42 | 44 | No paint board | 770 |
| Experiment 2 | 28 October | 6.2 | 0.29 | 24.3 | 27.8 | 39 | 45 | 0% $TiO_2$ | 1063 |
| Experiment 3 | 29 October | 7.9 | 0.19 | 21.4 | 27.1 | 44 | 49 | 3.5% $TiO_2$ | 2911 |





**Table 4.** List of the RACM2 compounds initialized. Definition, carbon valence and concentrations at the start of the experiments in $\mu$g.m$^{-3}$. Compounds marked with a symbol (*) were not measured experimentally, but were estimated based on simulations, so as to assess their importance regarding initial conditions (see section 3.1).

| Species | Definition | Carbon # | Exp1 | Exp2 | Exp3 |
|---|---|---|---|---|---|
| | *Organic compounds* | | | | |
| ACD | Acetaldehyde | 2 | 15.78 | 30.16 | 39.95 |
| ACT | Acetone | 3 | 11.77 | 18.80 | 18.30 |
| ALD | C3 and higher aldehydes | 3 | 22.18 | 63.31 | 107.0 |
| BALD | Benzaldehyde and other aromatic aldehydes | 7 | 0.559 | 2.437 | 5.879 |
| BEN | Benzene | 6 | 1.220 | 2.127 | 2.203 |
| CO | Carbon monoxide (*) | 1 | 0.535 | 0.517 | 3.581 |
| CSL | Cresol and other hydroxy substituted aromatics | 7 | 0.010 | 0.129 | 0.239 |
| DCB1 | Unsaturated dicarbonyls (*) | 4.5 | 0.238 | 0.336 | 0.888 |
| DCB2 | Unsaturated dicarbonyls (*) | 7 | 0.375 | 0.530 | 1.403 |
| DCB3 | Unsaturated dicarbonyls (*) | 4 | 0.452 | 0.643 | 1.849 |
| HCHO | Formaldehyde | 1 | 34.88 | 42.88 | 51.58 |
| ISO | Isoprene | 5 | 0.314 | 0.747 | 1.160 |
| KET | Ketones | 5 | 3.002 | 8.503 | 11.50 |
| LIM | d-limonene and other cyclic diene-terpenes | 10 | 6.151 | 8.799 | 8.273 |
| MACR | Methacrolein | 4 | 3.289 | 4.803 | 5.093 |
| MGLY | Peroxy radicals formed from MEK (*) | 3 | 0.355 | 0.415 | 1.640 |
| MOH | Methanol | 1 | 15.29 | 25.90 | 38.72 |
| OLI | Internal alkenes | 5 | 2.910 | 7.802 | 15.55 |
| OLT | Terminal alkenes | 3.8 | 20.20 | 35.33 | 62.52 |
| ORA1 | Formic acid | 1 | 30.52 | 33.10 | 30.74 |
| ORA2 | Acetic acid and higher acids | 2 | 74.17 | 123.1 | 168.1 |
| PAN | Peroxyacetyl nitrate and higher saturated PANs (*) | 2 | 0.360 | 0.107 | 1.511 |
| PHEN | Phenol | 6 | 0.554 | 0.567 | 0.585 |
| PPN | Peroxypropionyl nitrate (*) | 3 | 0.483 | 0.195 | 3.978 |
| ROH | C3 and higher alcohols | 3 | 10.32 | 43.99 | 105.0 |
| UALD | Unsaturated aldehydes | 5 | 0.687 | 1.385 | 1.968 |
| TOL | Toluene and less reactive aromatics | 7.1 | 3.580 | 4.702 | 4.684 |
| XYL | Xylene and less reactive aromatics | 8.9 | 31.37 | 62.72 | 38.48 |
| | *Inorganic compounds* | | | | |
| HONO | Nitrous acid | | 2.706 | 4.560 | 2.384 |
| NO | Nitric oxide | | 1.344 | 5.352 | 3.226 |
| NO2 | Nitrogen dioxide | | 12.02 | 5.087 | 9.184 |
| O3 | Ozone | | 2.767 | 0.000 | 0.992 |





**Table 5.** Parameter values for each type of simulation and each experiment. The parameters are : filtration factor [-], speed of air [m.s$^{-1}$], NO$_2$ hydrolysis stoichiometric coefficients $\gamma_i$ [-], uptake coefficients $\gamma_i$ [-], surface reactions kinetic rates $k_S$ [s$^{-1}$], desorption reactions kinetic rates $k'_{i,\,(ad)}$ [s$^{-1}$.mlc$^{-1}$]. Set 1 refers to the simulations with optimized parameters, Set 2 refers to the simulations with common $k_S$ constants, Set 3 refers to the simulations with common parameter values. The factor $f$ is allowed to vary between experiments; for the rest of the parameters, all of the values that differ from the Set 3 solution are denoted in bold.

| Set of parameters | Set 1 | | | Set 2 | | | Set 3 | | |
|---|---|---|---|---|---|---|---|---|---|
| Experiment | Exp 1 | Exp 2 | Exp3 | Exp 1 | Exp 2 | Exp3 | Exp 1 | Exp 2 | Exp3 |
| $f$ | 0.25 | 0.33 | 0.30 | 0.25 | 0.33 | 0.30 | 0.25 | 0.33 | 0.30 |
| $u_{\text{inf}}$ | 0.24 | 0.24 | 0.24 | 0.24 | 0.24 | 0.24 | 0.24 | 0.24 | 0.24 |
| $\beta_{\text{HNO}_3}$ | 0.47 | 0.47 | 0.47 | 0.47 | 0.47 | 0.47 | 0.47 | 0.47 | 0.47 |
| $\beta_{\text{HONO}}$ | 0.53 | 0.53 | 0.53 | 0.53 | 0.53 | 0.53 | 0.53 | 0.53 | 0.53 |
| $\gamma_{\text{NO}}$ | $8 \times 10^{-9}$ | $8 \times 10^{-9}$ | $8 \times 10^{-9}$ | $8 \times 10^{-9}$ | $8 \times 10^{-9}$ | $8 \times 10^{-9}$ | $8 \times 10^{-9}$ | $8 \times 10^{-9}$ | $8 \times 10^{-9}$ |
| $\gamma_{\text{NO}_2}$ | **$1.5 \times 10^{-6}$** | $5 \times 10^{-6}$ | $5 \times 10^{-6}$ | $5 \times 10^{-6}$ | $5 \times 10^{-6}$ | $5 \times 10^{-6}$ | $5 \times 10^{-6}$ | $5 \times 10^{-6}$ | $5 \times 10^{-6}$ |
| $\gamma_{\text{HONO}}$ | **$7 \times 10^{-7}$** | $2 \times 10^{-8}$ | $2 \times 10^{-8}$ | $2 \times 10^{-8}$ | $2 \times 10^{-8}$ | $2 \times 10^{-8}$ | $2 \times 10^{-8}$ | $2 \times 10^{-8}$ | $2 \times 10^{-8}$ |
| $\gamma_{\text{O}_3}$ | $1 \times 10^{-6}$ | $1 \times 10^{-6}$ | $1 \times 10^{-6}$ | $1 \times 10^{-6}$ | $1 \times 10^{-6}$ | $1 \times 10^{-6}$ | $1 \times 10^{-6}$ | $1 \times 10^{-6}$ | $1 \times 10^{-6}$ |
| $k_{\text{S1}}$ | **$5 \times 10^{-4}$** | $3 \times 10^{-3}$ | $3 \times 10^{-3}$ | $3 \times 10^{-3}$ | $3 \times 10^{-3}$ | $3 \times 10^{-3}$ | $3 \times 10^{-3}$ | $3 \times 10^{-3}$ | $3 \times 10^{-3}$ |
| $k_{\text{S2}}$ | $1 \times 10^{-14}$ | **$4 \times 10^{-14}$** | **$4 \times 10^{-13}$** | $1 \times 10^{-13}$ | $1 \times 10^{-13}$ | $1 \times 10^{-13}$ | $1 \times 10^{-13}$ | $1 \times 10^{-13}$ | $1 \times 10^{-13}$ |
| $k_{\text{S3}}$ | **$5 \times 10^{-5}$** | $1 \times 10^{-5}$ | $1 \times 10^{-5}$ | $1 \times 10^{-5}$ | $1 \times 10^{-5}$ | $1 \times 10^{-5}$ | $1 \times 10^{-5}$ | $1 \times 10^{-5}$ | $1 \times 10^{-5}$ |
| $k_{\text{S4}}$ | $2 \times 10^{-15}$ | $2 \times 10^{-15}$ | $2 \times 10^{-15}$ | $2 \times 10^{-15}$ | $2 \times 10^{-15}$ | $2 \times 10^{-15}$ | $2 \times 10^{-15}$ | $2 \times 10^{-15}$ | $2 \times 10^{-15}$ |
| $k'_{\text{NO (ad)}}$ | **$7 \times 10^{-23}$** | **$6 \times 10^{-21}$** | **$3 \times 10^{-20}$** | **$2 \times 10^{-22}$** | **$3 \times 10^{-21}$** | $8 \times 10^{-21}$ | **$2 \times 10^{-22}$** | $8 \times 10^{-21}$ | $8 \times 10^{-21}$ |
| $k'_{\text{NO}_2 \text{ (ad)}}$ | $1 \times 10^{-23}$ | $1 \times 10^{-22}$ | $1 \times 10^{-22}$ | $1 \times 10^{-22}$ | $1 \times 10^{-22}$ | $1 \times 10^{-22}$ | $1 \times 10^{-22}$ | $1 \times 10^{-22}$ | $1 \times 10^{-22}$ |
| $k'_{\text{HONO (ad)}}$ | **$2.5 \times 10^{-22}$** | **$4.5 \times 10^{-22}$** | **$1.5 \times 10^{-22}$** | **$1 \times 10^{-22}$** | **$7 \times 10^{-22}$** | **$1.2 \times 10^{-22}$** | $5 \times 10^{-22}$ | $5 \times 10^{-22}$ | $5 \times 10^{-22}$ |



Geoscientific Model Development Discussions — Open Access — EGU



**Table 6.** Simulations evaluations with respect to the four modelled inorganic compounds : simulated average, relative error between simulated and measured average, Root-Mean-Square Error (RMSE), Mean Normalized Gross Error (MNGE) and Mean Normalized Bias Error (MNBE).

| Experiment | Exp 1 | | | | | Exp 2 | | | | | Exp 3 | | | | |
| --- | --- | --- | --- | --- | --- | --- | --- | --- | --- | --- | --- | --- | --- | --- | --- |
| Species | $O_3$ | NO | $NO_2$ | HONO | NOx | $O_3$ | NO | $NO_2$ | HONO | NOx | $O_3$ | NO | $NO_2$ | HONO | NOx |
| Measured average [$\mu g.m^{-3}$] | 1.51 | 1.42 | 5.48 | 3.64 | 9.13 | 1.52 | 4.81 | 4.24 | 5.56 | 9.06 | 1.81 | 2.93 | 4.83 | 3.70 | 7.77 |
| **Set 1** | | | | | | | | | | | | | | | |
| Simulated average [$\mu g.m^{-3}$] | 0.73 | 0.96 | 2.65 | 3.54 | 3.62 | 0.65 | 4.36 | 4.57 | 5.52 | 8.94 | 0.73 | 2.07 | 5.88 | 3.64 | 7.96 |
| Relative error [%] | 51.65 | 32.39 | 51.64 | 2.74 | 60.35 | 57.23 | 9.35 | 7.78 | 0.71 | 1.32 | 59.66 | 29.35 | 21.73 | 1.62 | 2.44 |
| RMSE [$\mu g.m^{-3}$] | 1.01 | 0.58 | 3.49 | 0.12 | 5.68 | 1.80 | 0.98 | 0.95 | 0.15 | 0.45 | 1.45 | 0.98 | 1.26 | 0.16 | 0.47 |
| MNGE [%] | 49.15 | 36.65 | 45.27 | 2.84 | 61.13 | 56.68 | 16.03 | 21.33 | 1.09 | 4.12 | 57.97 | 33.35 | 28.91 | 3.42 | 5.32 |
| MNBE [%] | -47.88 | -34.81 | -45.27 | -2.74 | -61.13 | -26.99 | -9.07 | 11.40 | -0.07 | -1.33 | -52.79 | -31.52 | 26.59 | -0.38 | 2.55 |
| **Set 2** | | | | | | | | | | | | | | | |
| Simulated average [$\mu g.m^{-3}$] | 0.65 | 1.35 | 2.14 | 3.32 | 3.50 | 0.67 | 4.16 | 4.50 | 5.50 | 8.66 | 0.73 | 2.01 | 5.84 | 3.58 | 7.86 |
| Relative error [%] | 56.95 | 4.92 | 60.94 | 8.79 | 61.66 | 55.92 | 13.51 | 6.13 | 1.07 | 4.41 | 59.66 | 31.39 | 20.91 | 3.24 | 1.15 |
| RMSE [$\mu g.m^{-3}$] | 1.07 | 0.56 | 4.10 | 0.34 | 5.80 | 1.78 | 1.11 | 0.89 | 0.20 | 0.60 | 1.45 | 1.01 | 1.25 | 0.16 | 0.58 |
| MNGE [%] | 54.59 | 36.69 | 53.31 | 8.94 | 62.04 | 56.02 | 18.00 | 20.11 | 2.47 | 5.93 | 57.77 | 33.66 | 28.76 | 3.14 | 6.25 |
| MNBE [%] | -53.79 | -9.46 | -53.31 | -8.94 | -62.04 | -24.82 | -13.53 | 9.54 | -0.67 | -4.48 | -52.59 | -32.82 | 26.08 | -1.92 | 1.74 |
| **Set 3** | | | | | | | | | | | | | | | |
| Simulated average [$\mu g.m^{-3}$] | 0.65 | 1.37 | 2.08 | 7.11 | 3.46 | 0.61 | 4.74 | 4.70 | 5.04 | 9.45 | 0.75 | 1.93 | 5.63 | 6.56 | 7.56 |
| Relative error [%] | 56.95 | 3.52 | 62.04 | 95.32 | 62.10 | 59.86 | 1.45 | 10.84 | 9.35 | 4.30 | 58.56 | 34.12 | 16.56 | 77.29 | 2.70 |
| RMSE [$\mu g.m^{-3}$] | 1.08 | 0.58 | 4.13 | 3.71 | 5.82 | 1.83 | 0.88 | 1.05 | 0.57 | 0.69 | 1.43 | 1.11 | 1.02 | 3.09 | 0.46 |
| MNGE [%] | 54.88 | 38.10 | 54.93 | 92.88 | 62.63 | 58.70 | 16.96 | 23.44 | 8.24 | 6.39 | 57.18 | 38.45 | 22.54 | 78.34 | 5.15 |
| MNBE [%] | -54.09 | -8.83 | -54.93 | 92.88 | -62.63 | -30.44 | -0.20 | 14.48 | -8.23 | 4.41 | -51.48 | -36.82 | 20.15 | 78.29 | -3.27 |



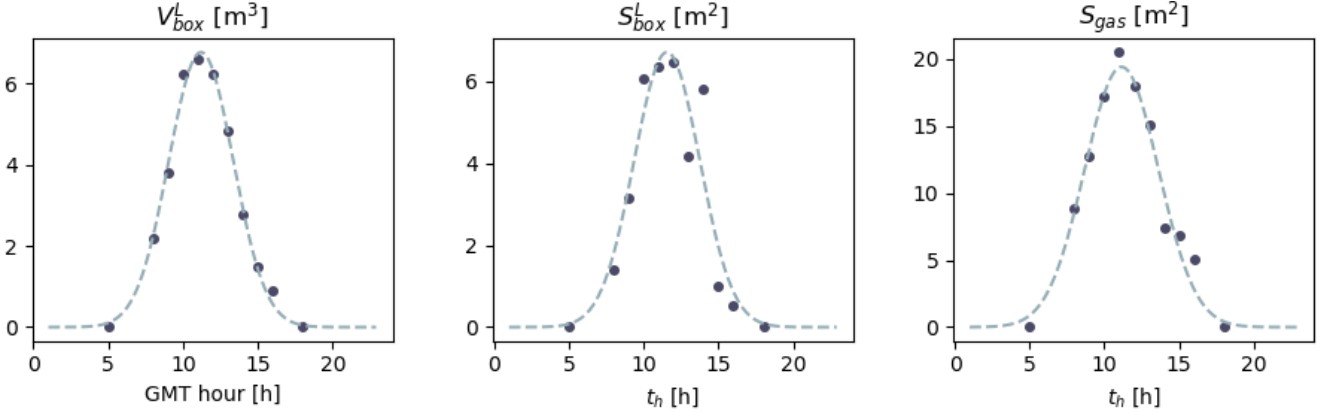

**Figure 1.** From the left to the right : evolution of $V_{\mathrm{box}}^{L}$, $S_{\mathrm{box}}^{L}$ and $S_{\mathrm{gas}}$ with GMT hour. The solid circles denote the values estimated numerically, the dashed lines are the gaussian laws they allow to infer. These parameterizations are representative of the time period ($27^{th}$ to $31^{th}$ October) and location (Martigues, France) of the experiments.





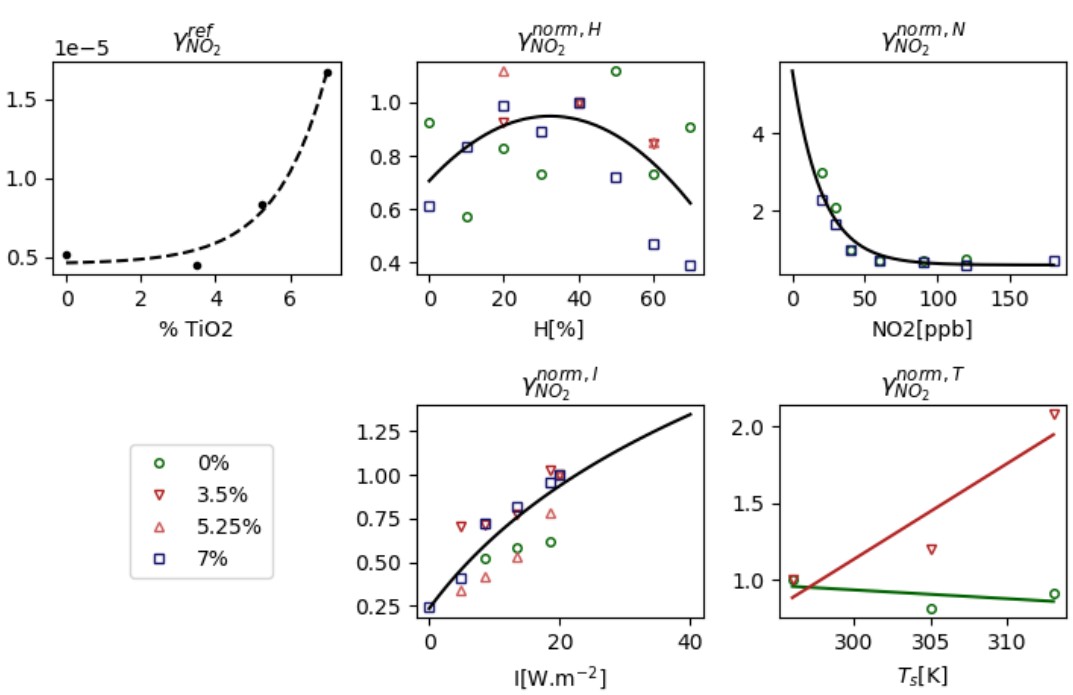

**Figure 2.** Evolution of $\gamma_{NO_2}$. The dots denote measurements, the open symbols denote normalized measurements (see text for details). The solid lines denote the parameterization as a function of $H$, $N$, $I$ and $T_s$ (Eqs. 25-28).



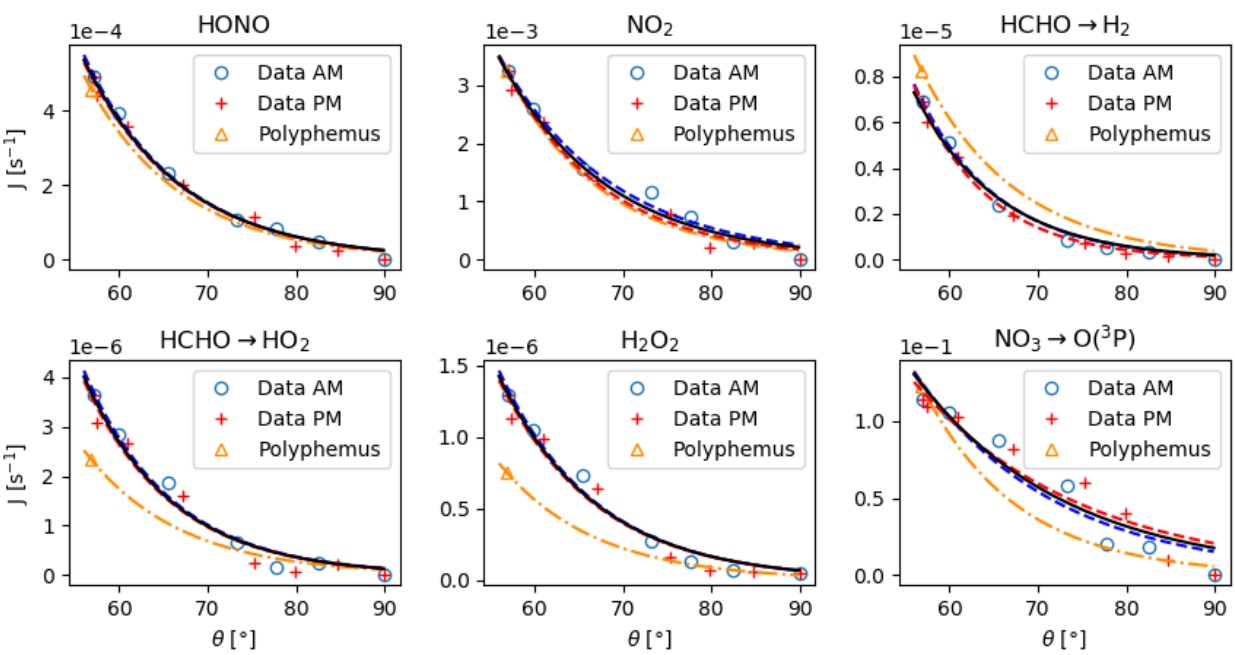

**Figure 3.** Photolysis rates as a function of zenith angle. The symbols '○' and '+' denote the experimental rates acquired on the $30^{th}$ October, in the morning and in the afternoon. The blue and red dashed lines are their parameterization using Eq. (31). The black solid line is the curve obtained by fitting both the data of the morning and those of the afternoon. The symbols '△' are the photolysis rates calculated with Eq. (30) with the cross sections and quantum yields taken from Polyphemus. The yellow dash-dotted line is the deriving evolution of $J$ with $\theta$ according to Eq. (33).



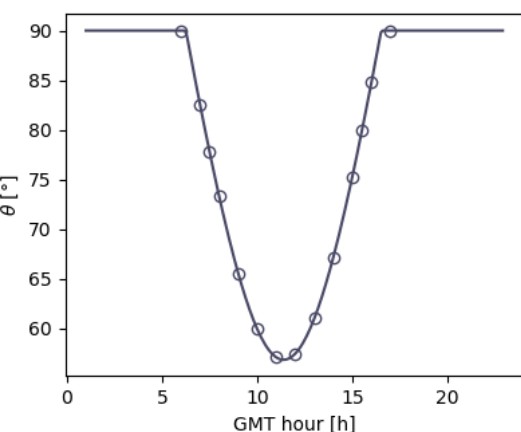

**Figure 4.** Zenith angle $\theta$ as a function of day hour, on the $27^{th}$ October at latitude $43.41°$ and longitude $5.06°$ (Martigues area). The '∘' symbols denote the hours of the $J_i$ measurements by the spectroradiometer.

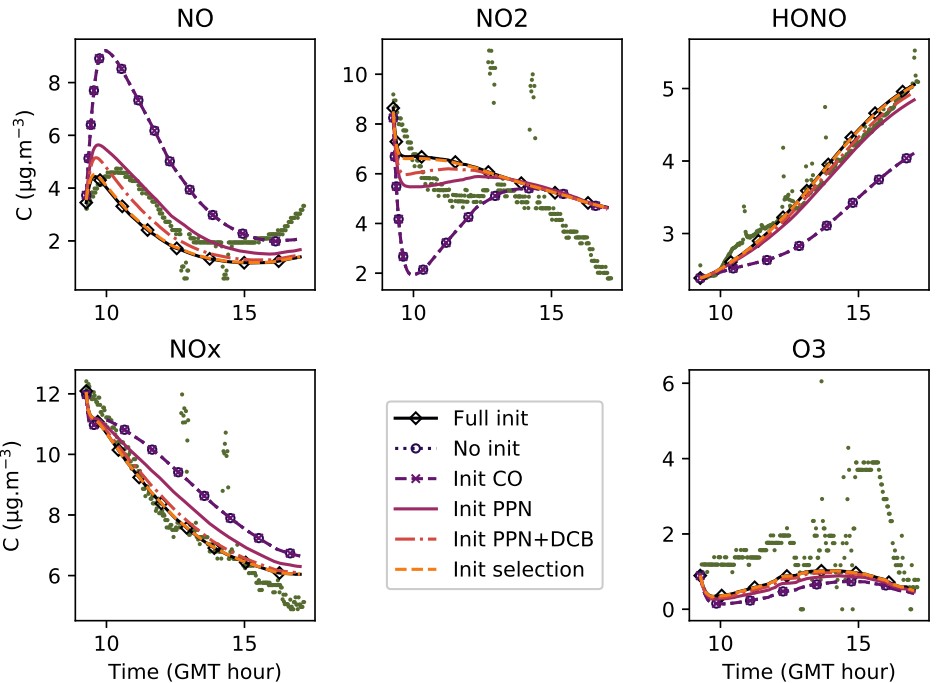

**Figure 5.** Inorganic concentration profiles for different initial conditions. The dots denote the experimental measurements (Experiment 3). "No init" means that all the compounds which were not detected during the campaign are a given a zero concentration at the start of the simulation. "Full init" means that all the compounds are initialized, even those which were not experimentally detected (see text for details). "Init CO" is like "No init" but with CO initialized. "Init PPN" is like "No init" but with PPN initialized . "Init PPN+DCB" is like "No init" but with PPN, DCB1, DCB2 and DCB3 initialized. "Init selection" is like "No init" but with PPN, DCB1, DCB2, DCB3, MGLY and PAN initialized. The curves corresponding to "Full init" and "Init selection" are close to be superimposed.





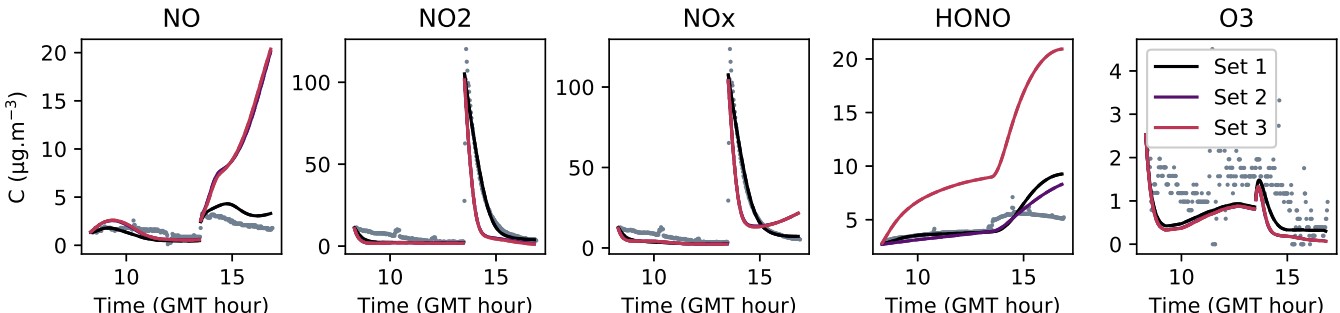

**Figure 6.** Simulation of Experiment 1 for the three sets of parameters. The dots denote the experimental records. The concentrations simulated in the sunlit box (solid lines) and the concentrations simulated in the shaded box (dashed lines) are superimposed.

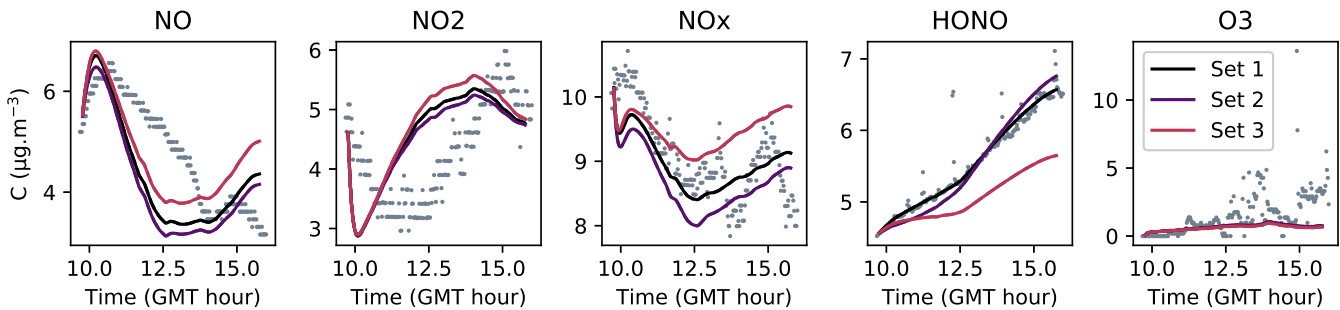

**Figure 7.** Simulation of Experiment 2 for the three sets of parameters. The dots denote the experimental records. The concentrations simulated in the sunlit box (solid lines) and the concentrations simulated in the shaded box (dashed lines) are superimposed.

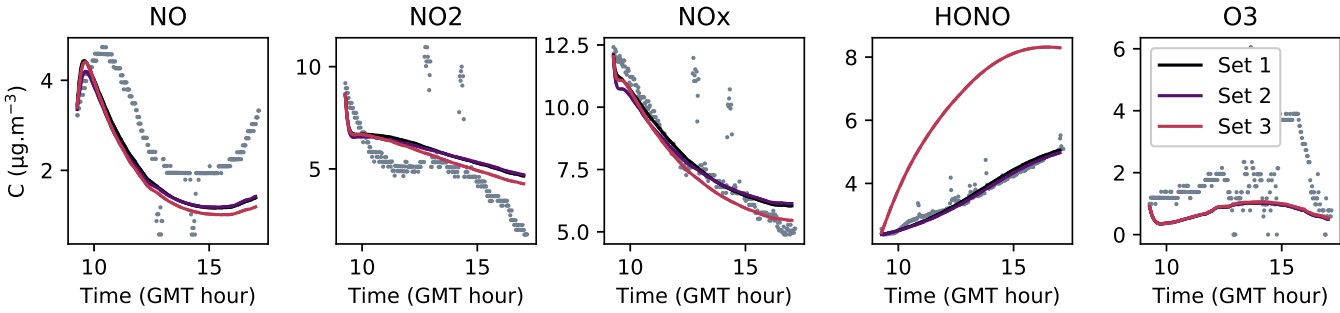

**Figure 8.** Simulation of Experiment 3 for the three sets of parameters. The dots denote the experimental records. The concentrations simulated in the sunlit box (solid lines) and the concentrations simulated in the shaded box (dashed lines) are superimposed.





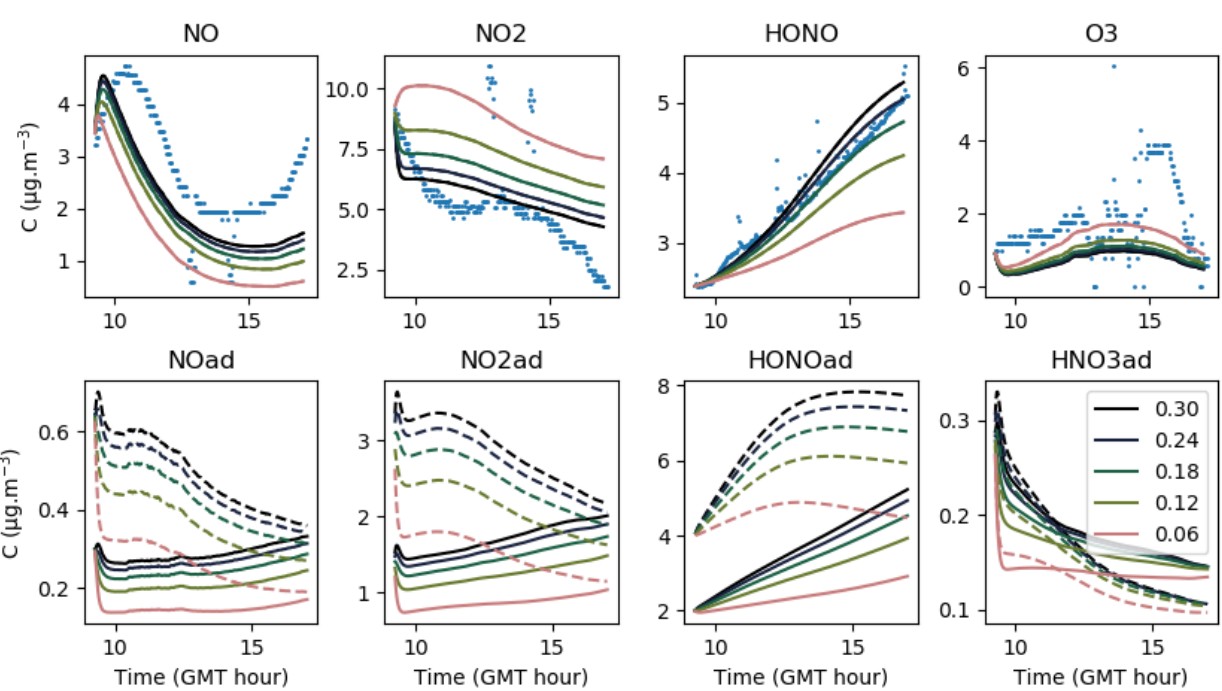

**Figure 9.** Gas-phase and adsorbed inorganic compounds simulated with different $u_{inf}$. The blue dots denote the experimental measurements (Experiment 3). The solid lines represent the concentrations in the sunlit box, the dashed lines the concentrations in the shaded box.



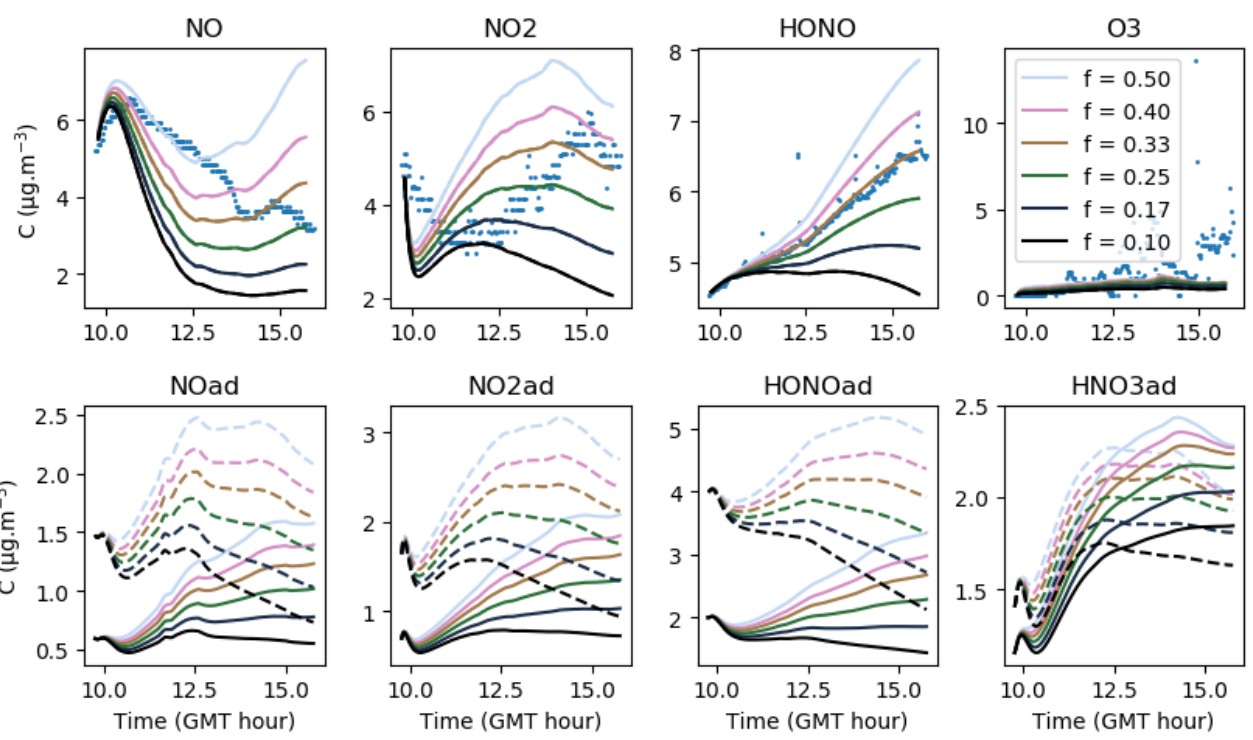

**Figure 10.** Gas-phase and adsorbed inorganic compounds simulated with different $f$. The blue dots denote the experimental measurements (Experiment 2). The solid lines represent the concentrations in the sunlit box, the dashed lines the concentrations in the shaded box.



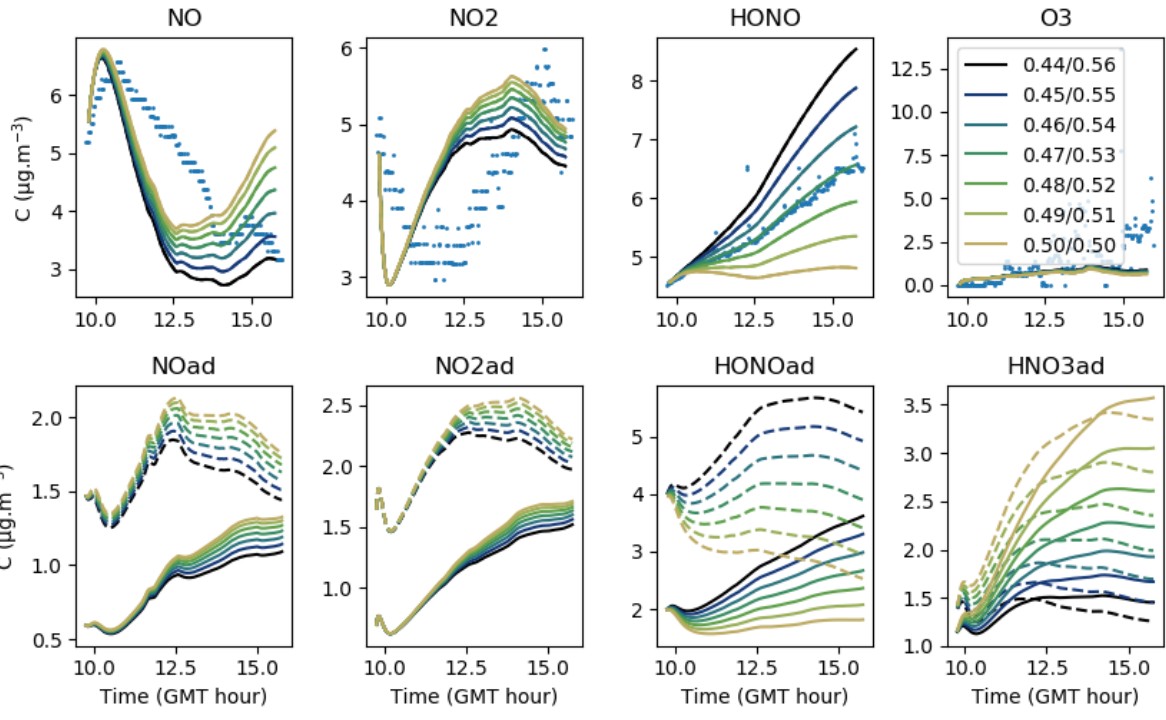

**Figure 11.** Gas-phase and adsorbed inorganic compounds simulated for different $\beta_{HNO_3}/\beta_{HONO}$, with $\beta_{HNO_3}/\beta_{HONO} < 1$. The blue dots denote the experimental measurements (Experiment 2). The solid lines represent the concentrations in the sunlit box, the dashed lines the concentrations in the shaded box.





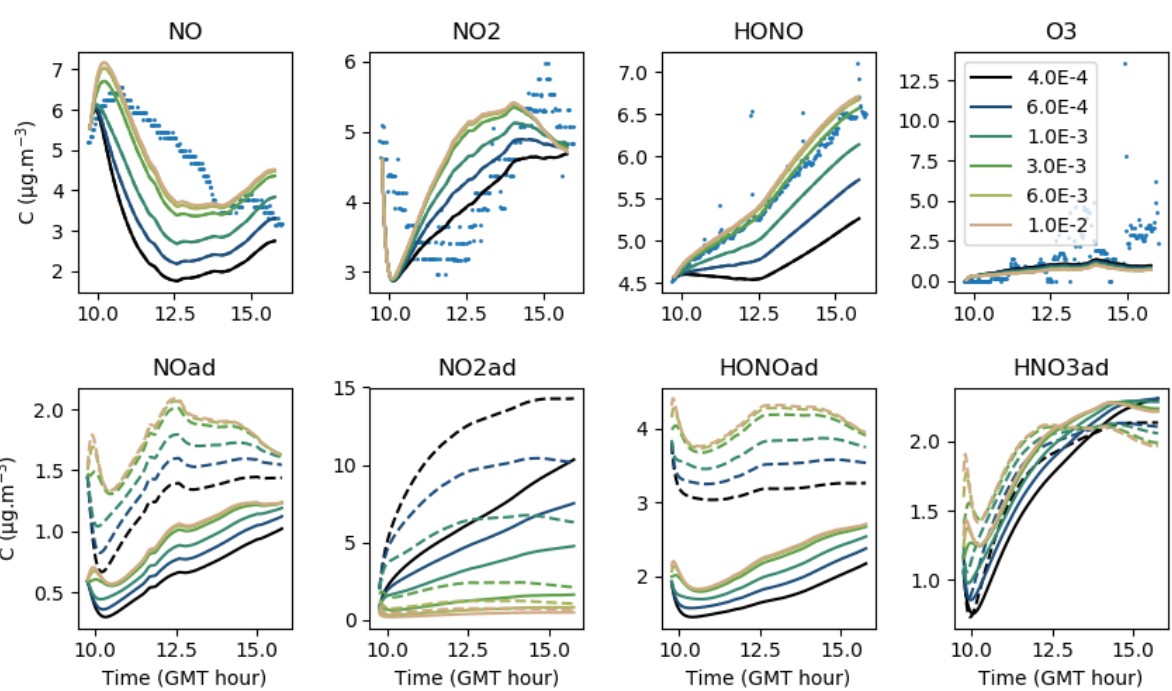

**Figure 12.** Gas-phase and adsorbed inorganic compounds simulated for different $k_{S1}$. The blue dots denote the experimental measurements (Experiment 2). The solid lines represent the concentrations in the sunlit box, the dashed lines the concentrations in the shaded box.

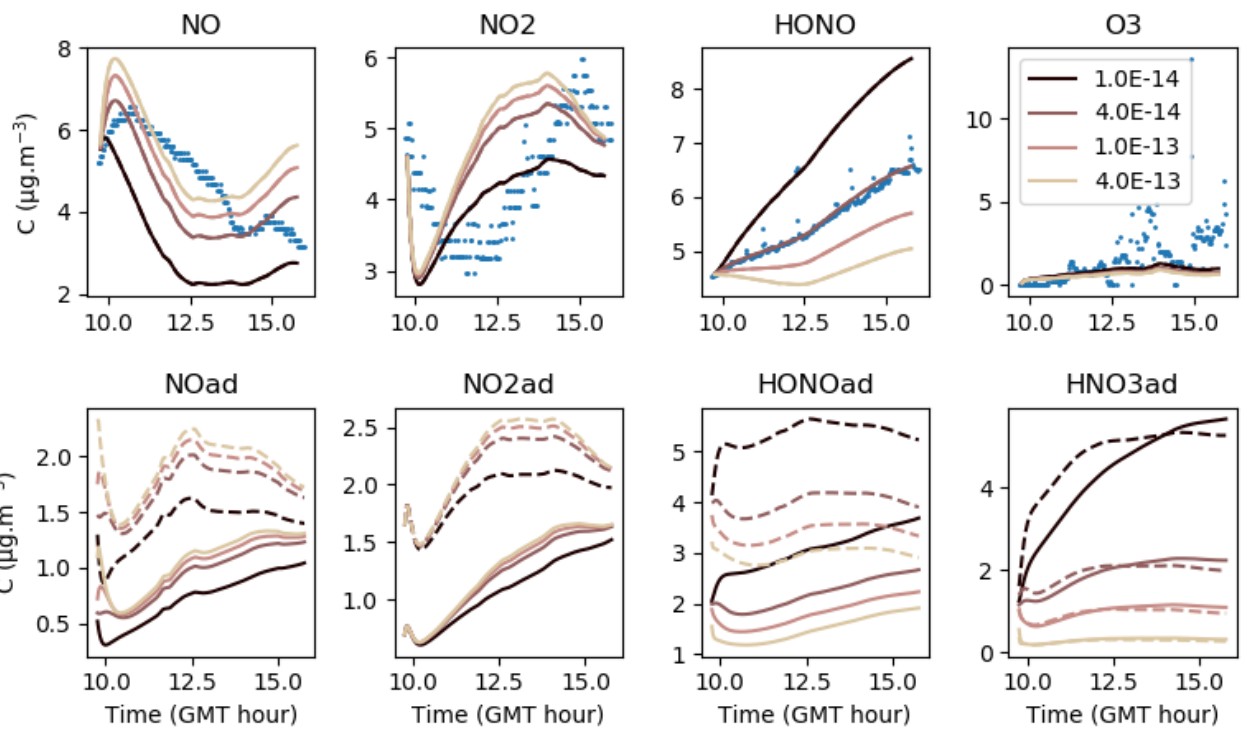

**Figure 13.** Gas-phase and adsorbed inorganic compounds simulated for different $k_{S2}$. The blue dots denote the experimental measurements (Experiment 2). The solid lines represent the concentrations in the sunlit box, the dashed lines the concentrations in the shaded box.



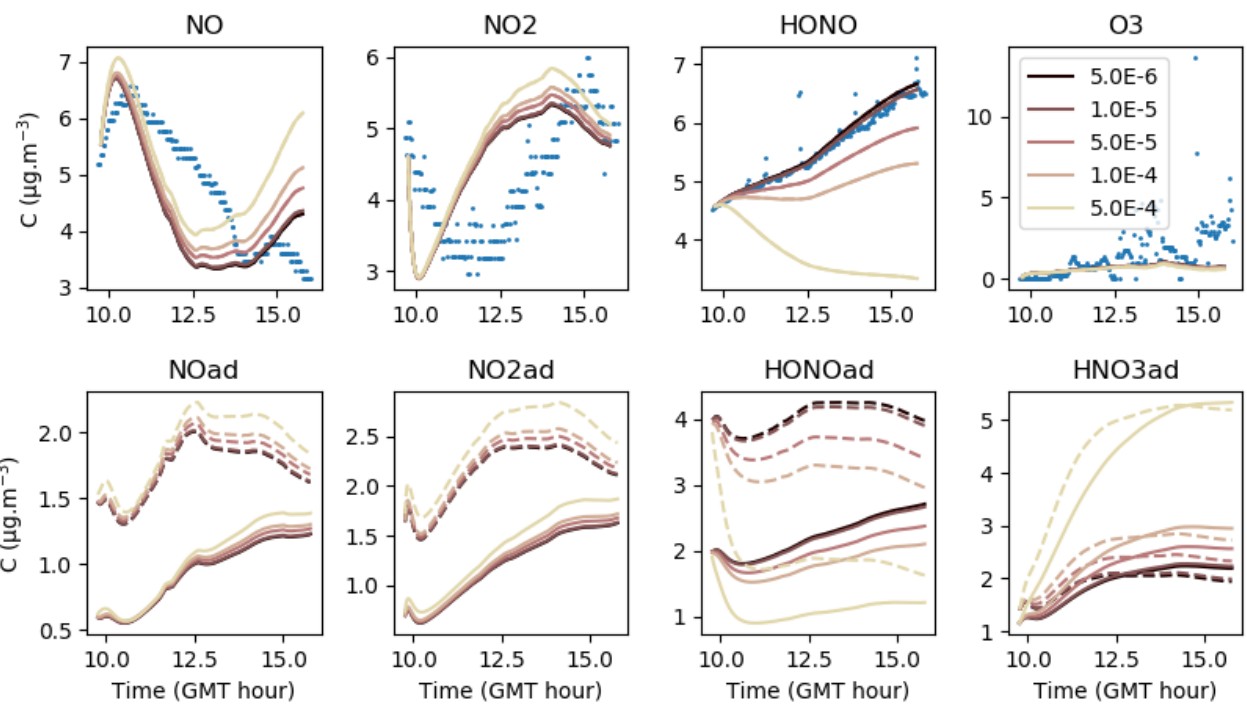

**Figure 14.** Gas-phase and adsorbed inorganic compounds simulated for different $k_{S3}$. The blue dots denote the experimental measurements (Experiment 2). The solid lines represent the concentrations in the sunlit box, the dashed lines the concentrations in the shaded box.



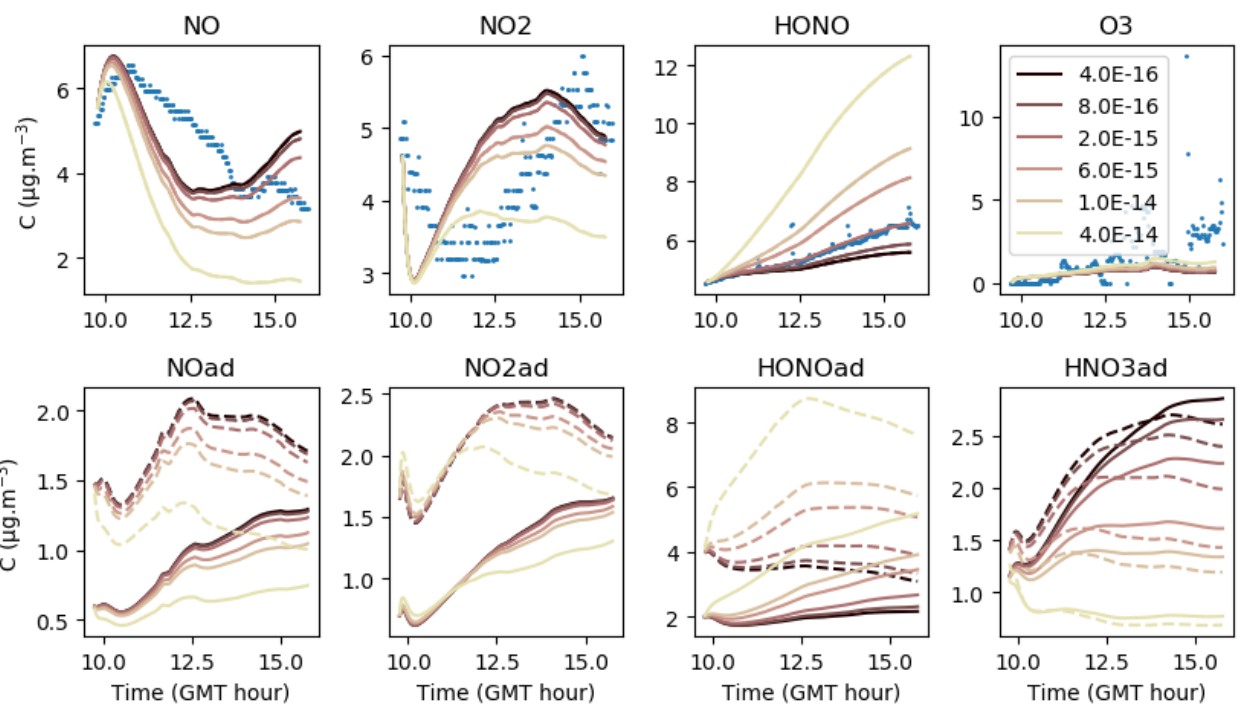

**Figure 15.** Gas-phase and adsorbed inorganic compounds simulated for different $k_{S4}$. The blue dots denote the experimental measurements (Experiment 2). The solid lines represent the concentrations in the sunlit box, the dashed lines the concentrations in the shaded box.



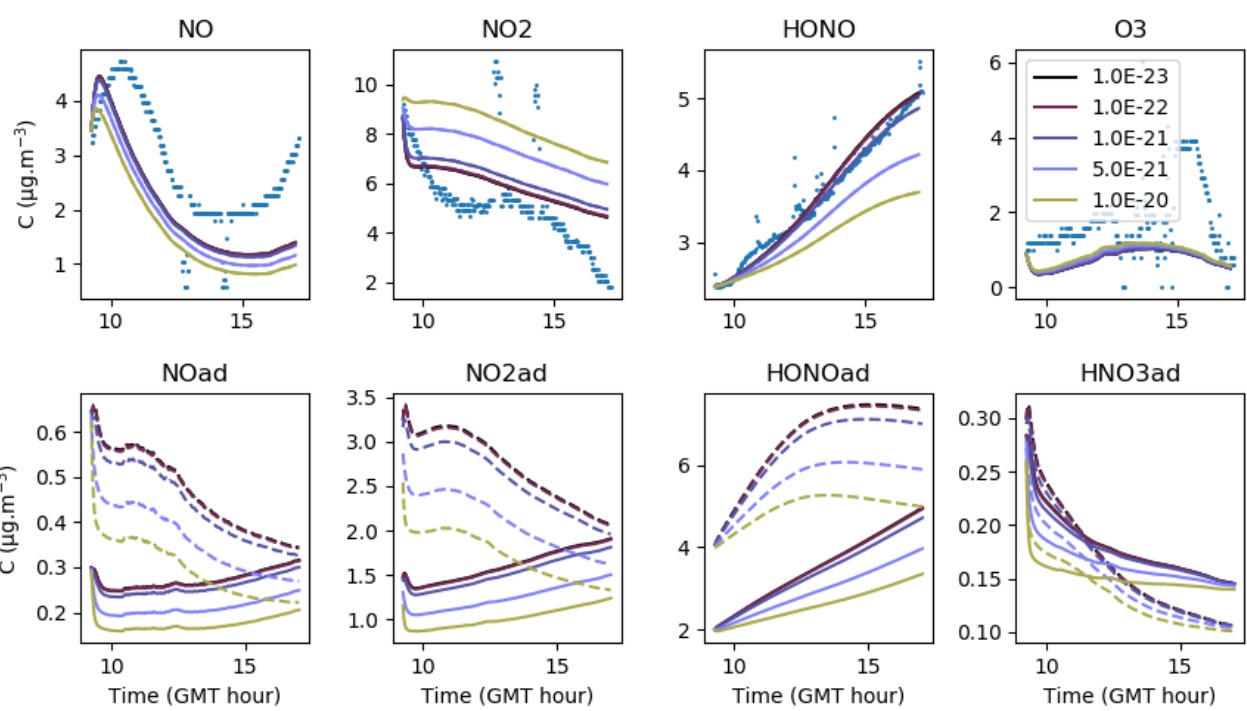

**Figure 16.** Gas-phase and adsorbed inorganic compounds simulated for different $k'_{NO_2 (ad)}$. The blue dots denote the experimental measurements (Experiment 3). The solid lines represent the concentrations in the sunlit box, the dashed lines the concentrations in the shaded box.



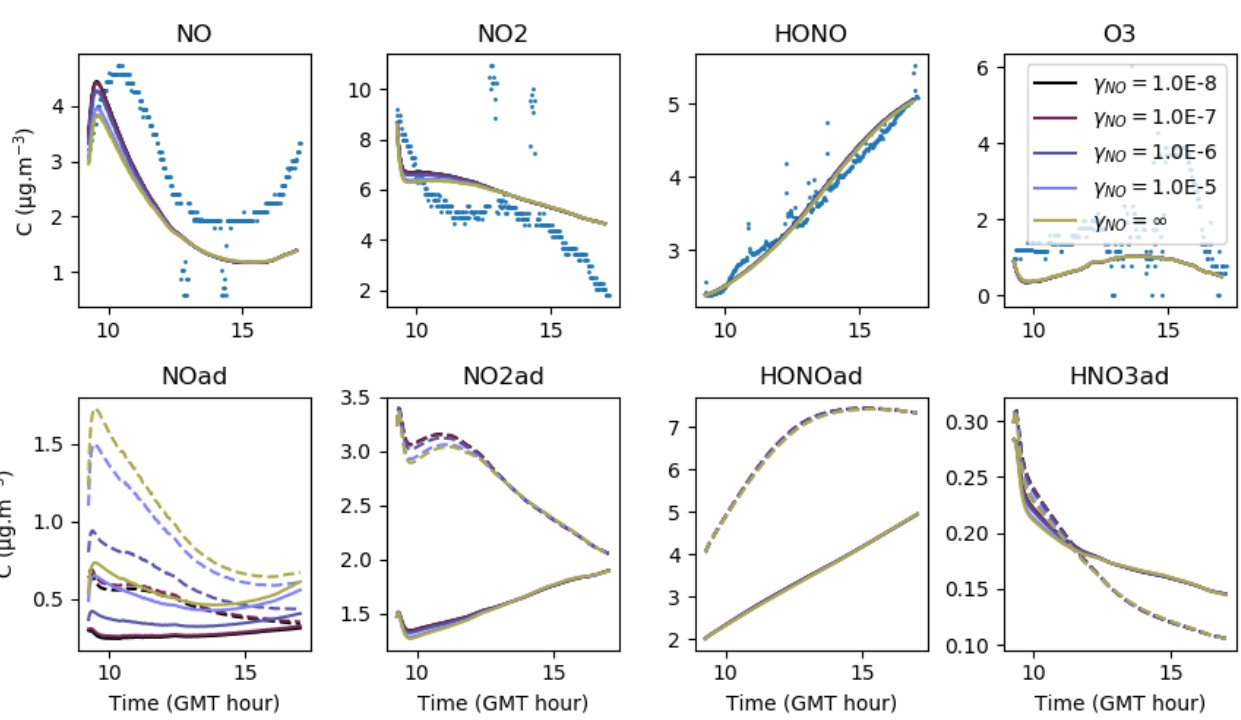

**Figure 17.** Gas-phase and adsorbed inorganic compounds simulated for different $\gamma_{NO}$. The blue dots denote the experimental measurements (Experiment 3). The solid lines represent the concentrations in the sunlit box, the dashed lines the concentrations in the shaded box.



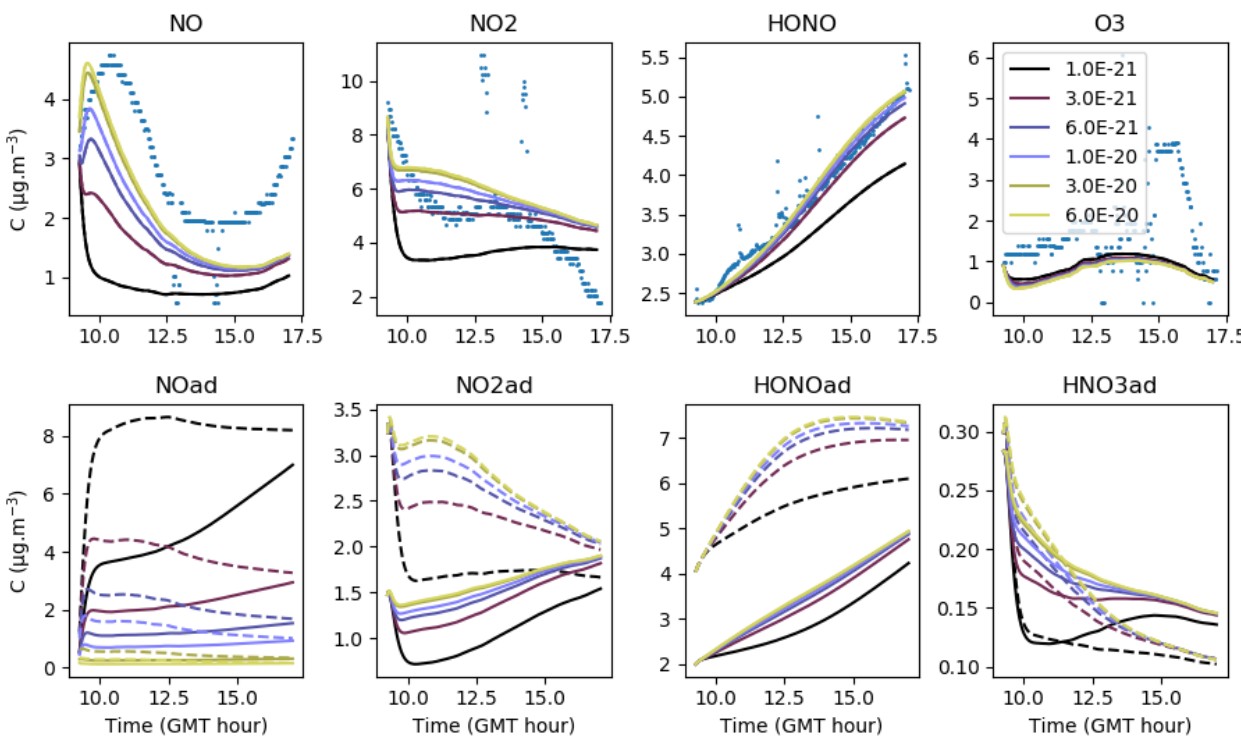

**Figure 18.** Gas-phase and adsorbed inorganic compounds simulated for different $k'_{NO\,(ad)}$. The blue dots denote the experimental measurements (Experiment 3). The solid lines represent the concentrations in the sunlit box, the dashed lines the concentrations in the shaded box.



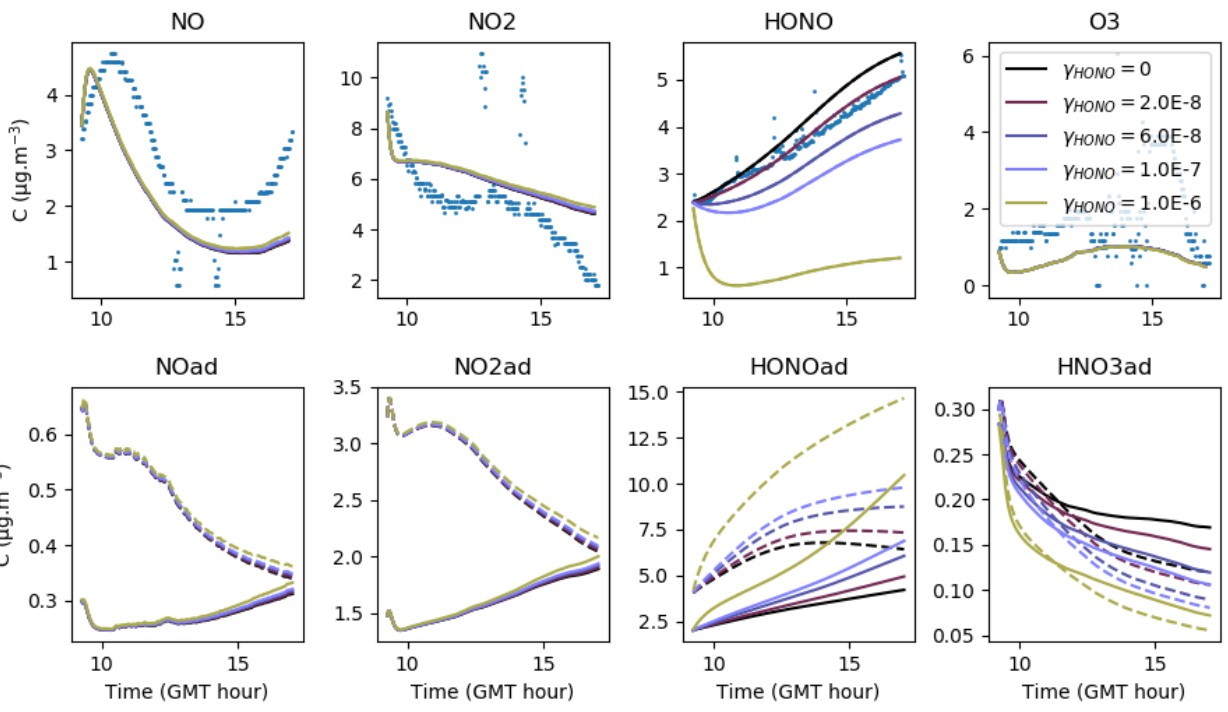

**Figure 19.** Gas-phase and adsorbed inorganic compounds simulated for different $\gamma_{HONO}$. The blue dots denote the experimental measurements (Experiment 3). The solid lines represent the concentrations in the sunlit box, the dashed lines the concentrations in the shaded box.



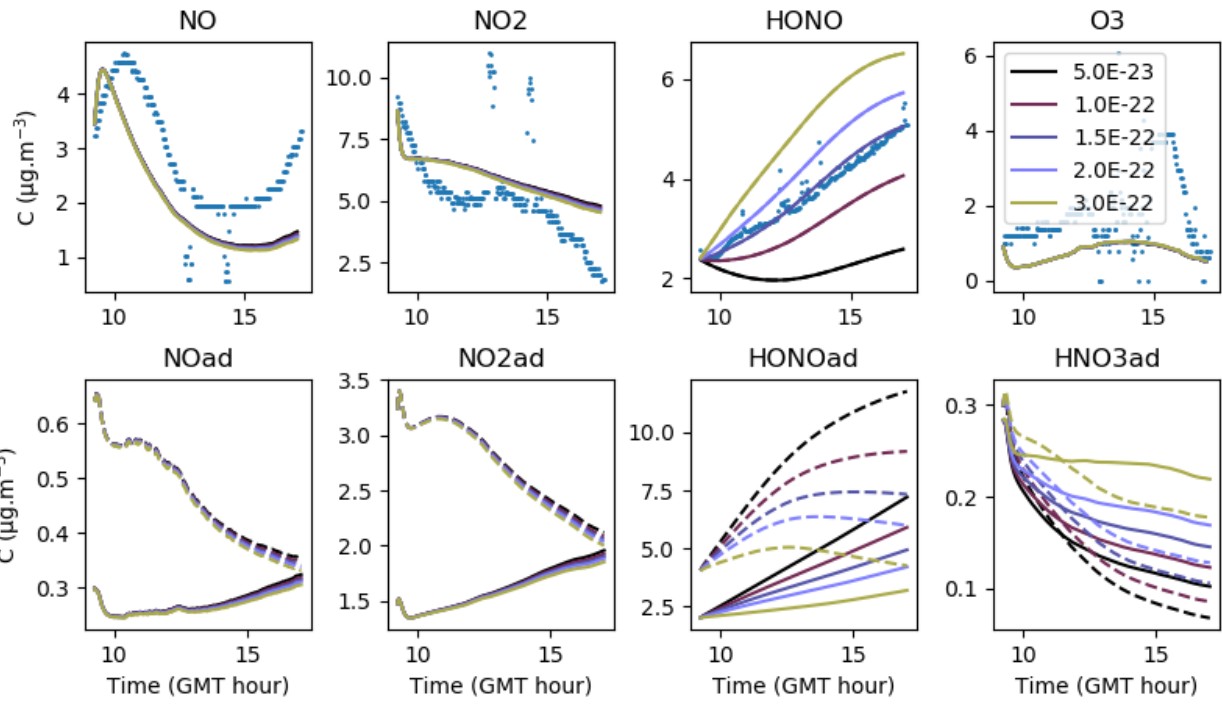

**Figure 20.** Gas-phase and adsorbed inorganic compounds simulated for different $k'_{HONO\,(ad)}$. The blue dots denote the experimental measurements (Experiment 3). The solid lines represent the concentrations in the sunlit box, the dashed lines the concentrations in the shaded box.



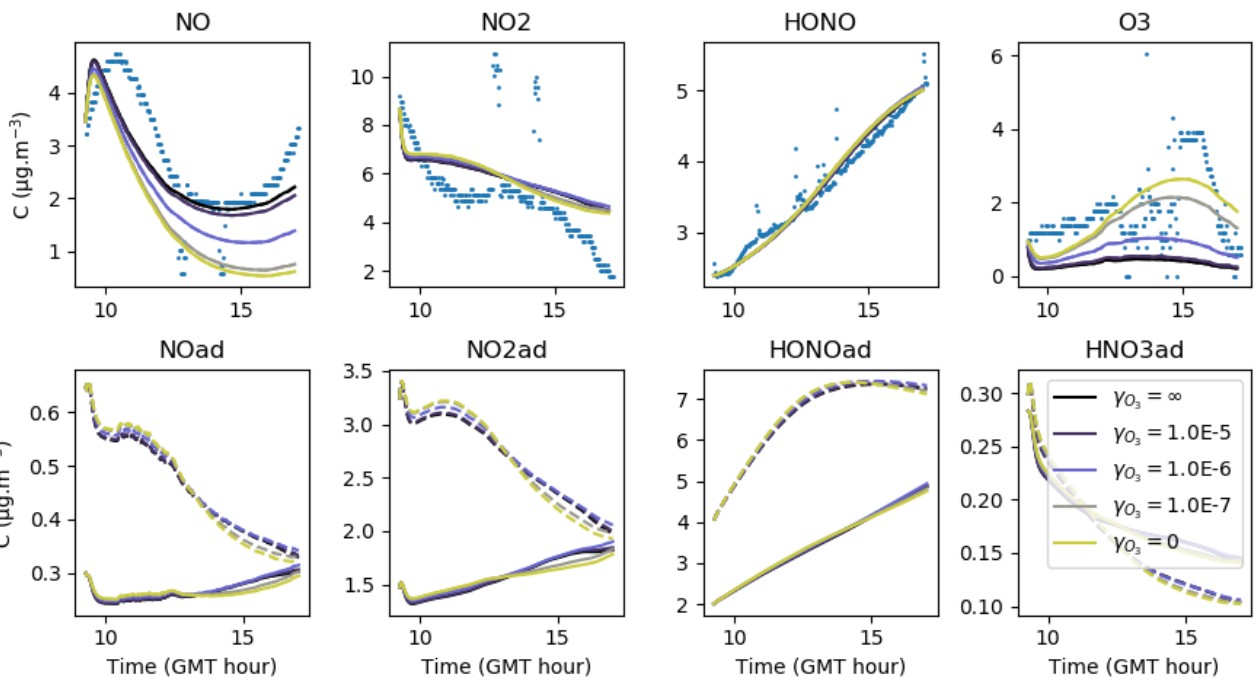

**Figure 21.** Gas-phase and adsorbed inorganic compounds simulated for different $\gamma_{O_3}$. The blue dots denote the experimental measurements (Experiment 3). The solid lines represent the concentrations in the sunlit box, the dashed lines the concentrations in the shaded box.



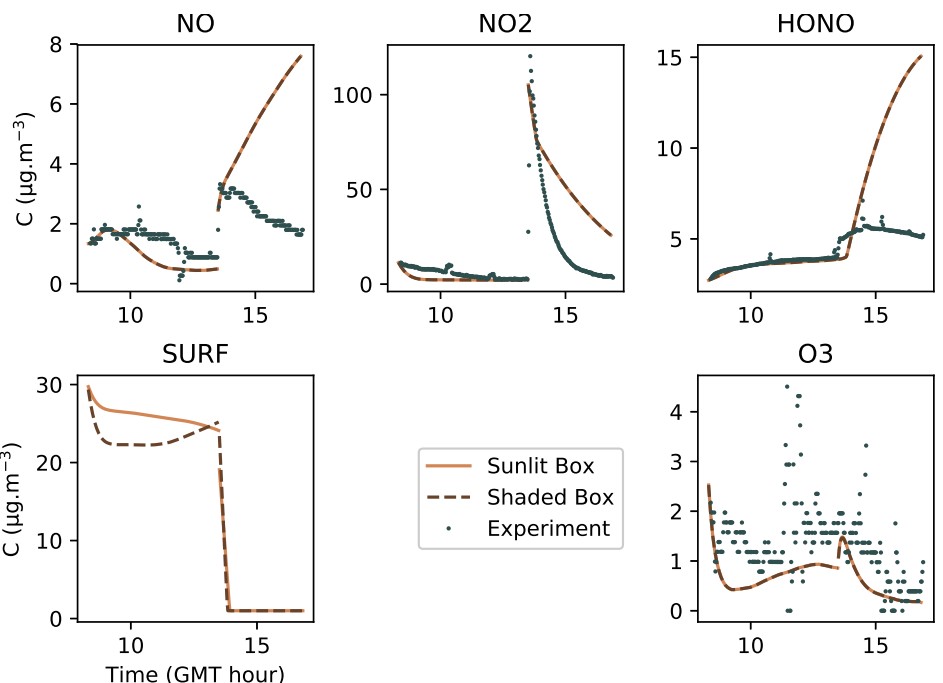

**Figure 22.** Implementation of a surface saturation effect using the compound SURF. The dots denote the experimental records, the solid lines denote the concentrations simulated in the sunlit box, the dashed lines denote the concentrations in the shaded box.





## Appendix A: Concentrations initialization

This appendix presents the figures corresponding to the initialization procedure described in section 3.3. In these figures, 'No init' is the simulation performed without initializing the compounds which were not detected during the campaign. 'Reference' is the simulation run corresponding to the "optimized simulation" (see section 4). The correspondence between these simulation runs and the experimental concentrations is assessed by computing the MNGE over the 5000 first seconds, denoted by the red dots, for NO and $NO_2$.

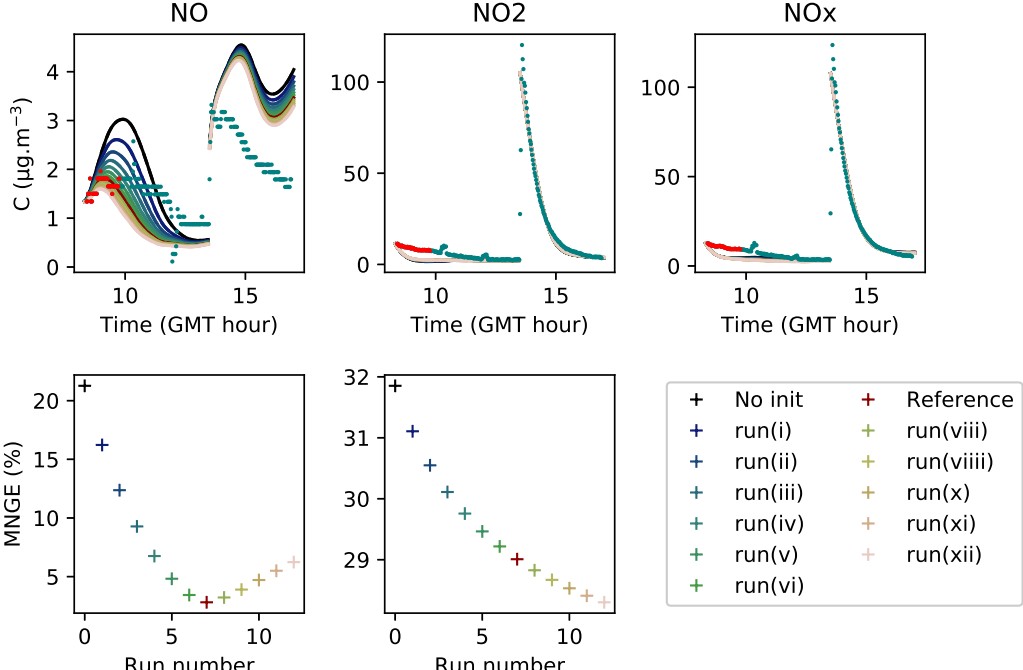

**Figure A1.** Concentrations initialization for Experiment 1 : for this experiment, proper initial concentrations are considered as achieved when the MNGE with respect to NO reaches a minimum. The MNGE of $NO_2$ remains high because the parameters optimization regarding $NO_2$ focused on the correct modelling of the second part of the experiment (see section 6, modelling high $NO_2$ concentrations).



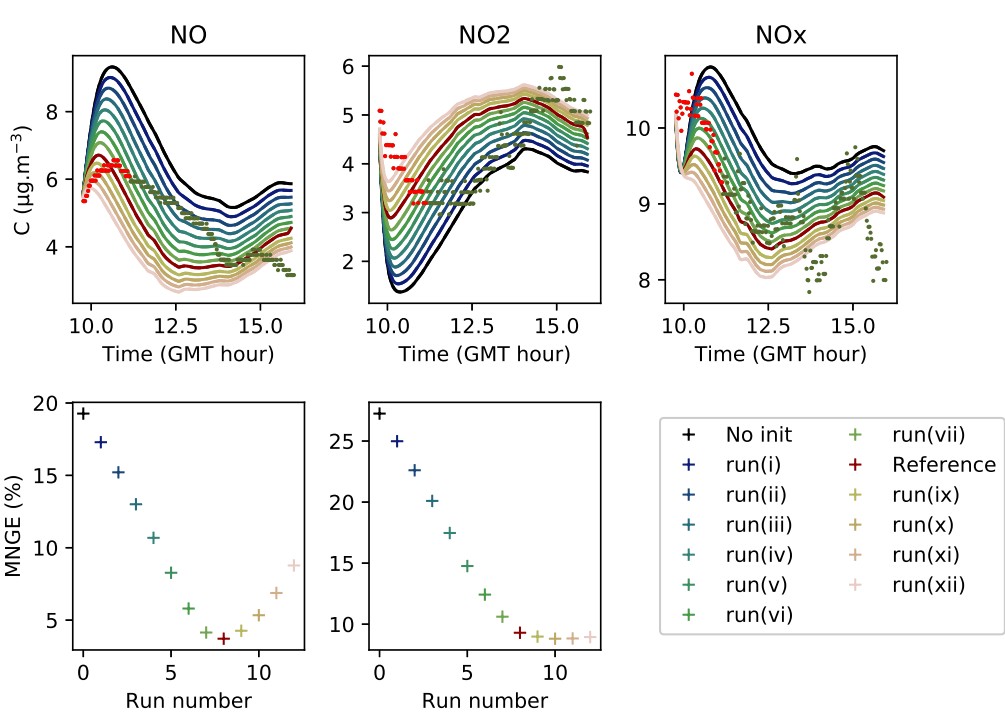

**Figure A2.** Concentrations initialization for Experiment 2 : for this experiment, proper initial concentrations are considered as achieved when the MNGE with respect to NO reaches a minimum, while the MNGE of $NO_2$ reaches stabilization.



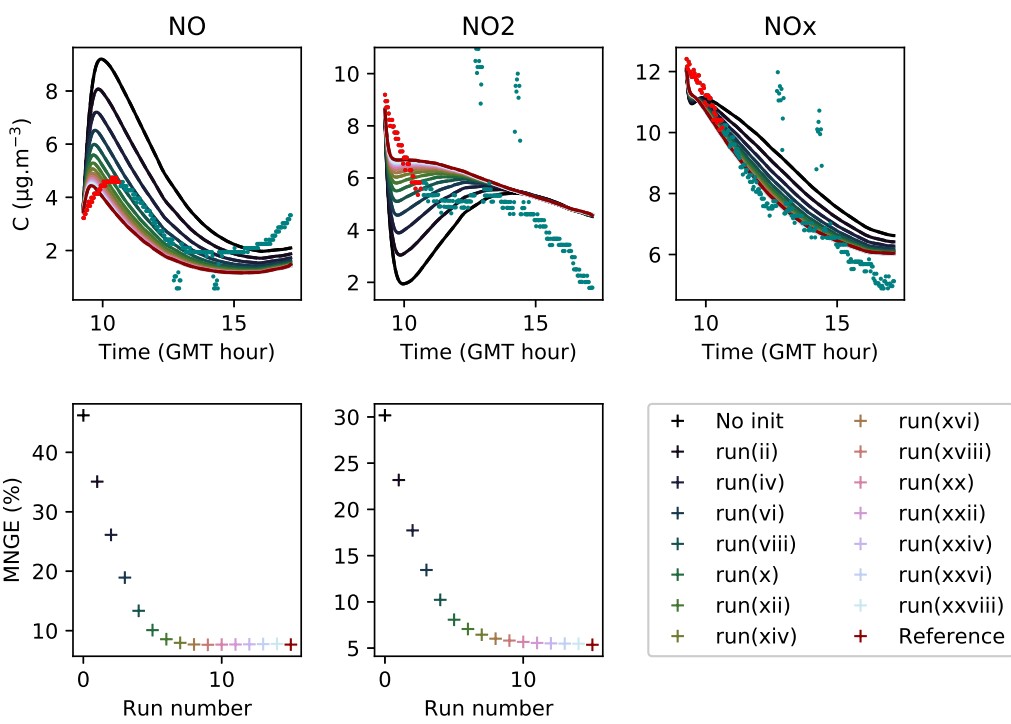

**Figure A3.** Concentrations initialization for Experiment 3 : for this experiment, proper initial concentrations are considered as achieved when the MNGE with respect to NO and NO₂ have converged to stable values.