# Peer review of "Combining homogeneous and heterogeneous chemistry to model inorganic compound concentrations in indoor environments: the $H^2I$ model (v1.0)"

_Geoscientific Model Development, 2020_

## Short Comment (SC1) · 14 Nov 2020

Dear authors,

in my role as Executive editor of GMD, I would like to bring to your attention our Editorial version 1.2:

https://www.geosci-model-dev.net/12/2215/2019/

This highlights some requirements of papers published in GMD, which is also available

on the GMD website in the 'Manuscript Types' section: http://www.geoscientific-model-development.net/submission/manuscript_types.html

In particular, please note that for your paper, the following requirement has not been met in the Discussions paper:

- Code must be published on a persistent public archive with a unique identifier for the exact model version described in the paper or uploaded to the supplement, unless this is impossible for reasons beyond the control of authors. All papers must include a section, at the end of the paper, entitled "Code availability". Here, either instructions for obtaining the code, or the reasons why the code is not available should be clearly stated. It is preferred for the code to be uploaded as a supplement or to be made available at a data repository with an associated DOI (digital object identifier) for the exact model version described in the paper. Alternatively, for established models, there may be an existing means of accessing the code through a particular system. In this case, there must exist a means of permanently accessing the precise model version described in the paper. In some cases, authors may prefer to put models on their own website, or to act as a point of contact for obtaining the code. Given the impermanence of websites and email addresses, this is not encouraged, and authors should consider improving the availability with a more permanent arrangement. Making code available through personal websites or via email contact to the authors is not sufficient. After the paper is accepted the model archive should be updated to include a link to the GMD paper.

As GMD requests persistent access to the exact version of the source code used for the model version presented in the paper and gitlab is not a persistent archive please provide for the presented source code a persistent identifier. As explained in https://www.geoscientific-model-development.net/about/manuscript_types.html the preferred reference to this release is through the use of a DOI which then can be cited

in the paper. For projects in GitHub a DOI for a released code version can easily be created using Zenodo, see https://guides.github.com/activities/citable-code/ for details. Finally note, that according to our new Editorial (v1.2) all data and analysis / plotting scripts should be made available.

Yours, Astrid Kerkweg

---

## Referee Comment (RC1) · Anonymous Referee #1 · 15 Feb 2021

The manuscript entitled "Combining homogeneous and heterogeneous chemistry to model inorganic compounds concentrations in indoor environments: the H2I model (v1.0)" by Fiorentino et al. describes a chemical box model that has been developed for indoor environments. There are relatively few thorough indoor air models. The manuscript focuses on the inclusion of heterogeneous chemistry, which is extremely important for indoor air. The development of this model represents an important new tool for the study of indoor air chemistry. While this is not the first example of an indoor model that treats heterogeneous chemistry, it is nonetheless an important one.

[Figure]

Overall, the paper is dense and model description is extensive. It is recommended that the authors consider including a summary table of symbols used, as the notation can be difficult to follow.

The most important conceptual difference that this reviewer has with the paper is the treatment of surface chemistry. A prior study by Mendez et al (Indoor Air, 2017) worked within a similar conceptual framework to the present study and was not able to achieve strong agreement between the model and measurements in some cases, particularly when NO2 was high. A similar deficiency is found in the present model output. Other frameworks, like treating the surface as a multilayer reservoir that has a 'depth' (so to speak) has been suggested by other authors (Collins et al., Environ. Sci. Technol. 2018). Models may be more able to tolerate high concentrations of NO2 using equilibrium, rather than strict competitive adsorption kinetics (i.e. - surface site saturation). Stronger consideration of this issue appears to make an appearance in the Discussion section as a possibility for future study. Further investigation of the possibility that competitive adsorption/desorption may not represent indoor surface films is strongly recommended.

The paper should be acceptable in Geoscientific Model Development after consideration of these reviewer comments and a minor revision.

Specific Comments: Line 98-100: "the first two-box model allowing to consider the evolution of light intensity of each part along the day, as well as the volumes they occupy" Other partially illuminated models exist, including recent reports that use CFD for studies of HONO chemistry in partially illuminated spaces. It would be worthwhile to explicitly place the present model in a broader context.

Line 103: "acquired in Martigues" Please be more specific.

Line 138: "S_j_paint is the surface of paint in the box j [m-2]" The units suggest that this is a surface area-to-volume ratio, but the text suggests that it might be a surface area. Please clarify or correct.

Line 158-159: "The complement of S_L_box to obtain the surface of the volume V_L_box is the same as the complement of S_S_box..." This wording is very confusing. Perhaps clarifying "surface area" and referring to the 'box' rather than 'the volume' would help.

Broader comment: Referring to various forms of "S" parameters as 'surfaces' appears in many locations. Please be more specific about the type of quantity that "S" denotes.

Line 188-189: Similar efforts, but with ozone, have been made by Morrison et al. (Indoor Air, 2019). https://doi.org/10.1111/ina.12601

Line 248-249: "deposition of the OH radical is considered negligible compared to its consumption by reactivity" Is this statement made in reference to the heterogeneous reactivity of OH or the homogeneous gas phase reactivity? Indoor multiphase reactivity by OH has been shown by Alwarda et al (Indoor Air, 2018).

Section 2.5: Have the authors considered a distance dependence to actinic flux? Dependence of actinic flux on wavelength, distance, and other variables have been characterized by Zhou et al. (Indoor Air, 2021) and Kowal et al. (Environ. Sci. Technol., 2017).

Line 394-395: The definition of filtration factor here seems different than this reviewer's previous understanding. The filtration factor (or penetration factor, in Sarwar et al (2002)) describes the ability for a specific pollutant to penetrate the building envelope. It is not necessarily a measure of how airtight a building might be. It is related to the route and loss processes involved in transporting a particular pollutant from outdoors to indoors, and is independent of the air exchange rate (which is controlled in part by the airtight-ness of the building). NOx penetration factors have been measured by Zhao et al. (Environ. Sci. Technol., 2019; doi: 10.1021/acs.est.9b02920). These authors found that the NO penetration factor was ~1 and for NO2 it was closer to 0.7 for the same environment. Please clarify and correct the text. Note: This reviewer clearly understands that perhaps filtration or penetration factors have not been determined

for the building being used in the present study, but citing values that are known for compounds in this study and that the filtration factor values are compound-specific are extremely important items for the reader to understand.

Line 393-395: This statement about competitive adsorption of H2O and HONO may not be relevant. Our current understanding of indoor surfaces includes the possibility that multilayers may exist. Please refer the overall comment on the treatment of surface chemistry at the top of this review.

Line 405-406: "no experimental validation of this production ratio is available" This is not true, as it has been a topic of research for many years. Most saliently, the group of Finlayson-Pitts has studied this concept, and the authors have cited several of their papers. In addition, Grassian and co-workers have studied this problem.

Section 5.1.2: Filtration factor – The description of the filtration factor is largely correct, but some of the description may confuse the meaning of the filtration factor.

Line 523: "outdoor pollutants like O3 and NO2". Why isn't NO in this list? NO mixing ratios outdoors are often greater than NO2 during daytime in cities when NO2 photolysis is rapid.

Line 533-534: Is it possible that the prevalence of Reaction S2 is overestimated?

Line 534: "As NO increases, NO2 increases by equilibration through homogeneous chemistry." This statement is unclear. It may be correct, but it is not clear enough to understand what it is supposed to mean.

Line 549-550: "The threshold value k_S1 = 0.003 s-1 is retained as it maximizes the NO2(ad) conversion." Why is this the most appropriate choice? Please further justify this decision. What impact does this decision have on other results?

Section 5.3.1: What happens if HONO and HNO3 dissociate?

Line 633: Please comment on how the uptake coefficient of HONO compares with the

literature.

Line 650-653: It is somewhat alarming to see that the parameters within the model must be altered in response to the reference experiments. The broader issue brought forth by Section 6 is discussed in the overall comments of this review and has been addressed by the authors.

Line 702: Nitric acid is most reactive with other radical species or reducing agents. It is not terribly reactive in the indoor environment – not close at all to the reactivity of OH. The small deposition velocity for NO comes from its slow reactivity with surface-bound materials. If there is more information about the reactivity of NO indoors, please cite.

---

## Referee Comment (RC2) · Anonymous Referee #2 · 18 Feb 2021

Review

[Figure]

Interesting paper that presents a model that considers homogeneous and heterogeneous chemistry,

Some remarks and suggestions,

- The English must be improved in all parts of the paper,
- page 1, line 8, "The model succeeds in simulating correctly", what you mean by correctly, please to precise, e.g., give error values comparing simulations and experiments
- page 4, line 107 suggestion to replace "master equation" mass balance equation "
- page 4, line 110, I suggest presenting a simple schema that clearly shows the domains of L, S, the boundary of domains, and all flows and mass fluxes, this consideration will help to understand more the model
- page 5, line 121/Eq2, could you explain why you multiply miS by VboxL/VboxS / same question for line 123/Eq3
- page 5, line 133, ... in these experiments..., which experiments? please precise
- Page 5, ... emission of the building itself... not clear which emission? indoor emissions due to activities/appliances/furnitures?
- page 6, line 149-line 156, must present a schema that shows an example of surface projections, surface boundaries, beams with window the window position considered
- page 6, line 153, line 155, are these equations valid for each season? each window position, surface and characteristics? please to highlight the limit of these equations,
- page 12, line 339, what parameters are assigned for VOCs, not clear, VOCs concentrations?
- page 12, line 341, is slower? it seems it is lower, not clear
- page 14, line 388, how the other uptake values can be adjusted by a user? from values in the literature?
- page 25, line 534, suggestion to replace "replicate" by "to predict"
- page 25, line 735, please to review this phrase "For the first time, O3, HONO and NOx species are simulated all at once and compared to experimental records acquired in a real room". is it the first study that estimates experimentally O3, HONO and NOx concentrations in a real environment?
- page 25, line 737 and line 738, please review this phrase "It is also the first two-box model allowing to consider the variations of direct and indirect light throughout the day".please to more explicit about this consideration by making the link with the study

**Fig. 1.**

---

## Author Comment (AC1) · 3 Apr 2021

Dear Executive Editor,

The code is now available from a GitHub repository, and is now referenced with a DOI. As required, an example of data analysis/plotting script has been added to the repository.

Yours sincerely,

---

## Author Comment (AC2) · 3 Apr 2021

First of all, the authors thank the reviewers for their comments and suggestions which brought important improvements to this manuscript. The answers to the questions are detailed below.

**Anonymous Referee #1**

General Comments :

• Overall, the paper is dense and model description is extensive. It is recommended that the authors consider including a summary table of symbols used, as the notation can be difficult to follow.
>> A summary of symbols is now included in appendix.

• The most important conceptual difference that this reviewer has with the paper is the treatment of surface chemistry. A prior study by Mendez et al (Indoor Air, 2017) worked within a similar conceptual framework to the present study and was not able to achieve strong agreement between the model and measurements in some cases, particularly when $NO_2$ was high. A similar deficiency is found in the present model output. Other frameworks, like treating the surface as a multilayer reservoir that has a 'depth' (so to speak) has been suggested by other authors (Collins et al., Environ. Sci. Technol. 2018). Models may be more able to tolerate high concentrations of $NO_2$ using equilibrium, rather than strict competitive adsorption kinetics (i.e. - surface site saturation).
Stronger consideration of this issue appears to make an appearance in the Discussion section as a possibility for future study. Further investigation of the possibility that competitive adsorption/desorption may not represent indoor surface films is strongly recommended.
>> The following paragraphs have been included :

Section 6, line 694-704:
*"Conversely, when the SURF ``concentration'' is decreased, the sorption of $NO_2$ and related species is reduced, which generates an excess of gas-phase $NO_y$.*
*From Fig.23, we may conclude that the parameterization of desorption needs improvement, or that the contribution of $NO_2$ to the secondary formation of HONO and NO is overestimated. These results agree with the work of Collins et al. (2018), who performed indoor time-resolved measurements of HONO and $NO_2$ under both positive and negative perturbations. Their measurements indicated a weak correlation between the concentrations of both species. Similar behaviour was observed during the SURFin campaign (Alvarez et al., 2013), where the HONO concentration increased rapidly after ventilation periods and remained in a near steady-state despite $NO_2$ variations. The SURF compound introduced by Mendez et al. (2017) to model these data made it possible to decrease the coupling between both species. However the modelled HONO concentration still retained more sensitivity to $NO_2$ than what the measurements indicated. It appears from Fig.23 that extending the definition of SURF to all sorbed compounds introduced in this model does not improve the problem."*

Section 7, line 755-770:
*"In light of this, it can be inferred that a more sophisticated parameterization of the desorption reactions may be required to improve this model. Rather than a competitive adsorption/desorption kinetics, the adsorption/desorption phenomenon may be represented by an equilibrium approach, which depends on the mass ratio adsorbed/volatilised. Such improvement may also alleviate the problems observed after the $NO_2$ injection during Experiment 1. Similar to Mendez et al. (2017), we observe that the measured increase in HONO after the $NO_2$ injection is moderate, and similar to the models tested by Mendez et al. (2017), the H2I model overestimates this increase in HONO concentration. Collins et al. (2018) succeeded in simulating gas-phase HONO at high $NO_2$ level using persistent source and sink processes, with only a small contribution from $NO_2$ uptake. To explain the poor correlation between both species concentrations, Collins et al. (2018) suggested that the gas-phase HONO*

*could be in equilibrium with a precursor reservoir of nitrite and/or HONO dissolved in surface films or sorbed on surfaces. Thanks to the introduction of the compound HONO(ad), the current model includes this type of reservoir. Indeed, the HONO produced by reaction S1 first remains adsorbed before being released to the gas-phase. However, even considering an intermediate compound HONO(ad), the connection between NO2 and HONO remains too strong. Different type of desorption model based on equilibrium rather than competition for surface sites may decrease this coupling. In addition, introducing another compound interacting with HONO(ad), such as nitrite, should further decrease the influence of reaction S1 on the HONO(ad) concentration, and thus, further attenuate the dependence of the gas-phase HONO on NO2."*

Conclusion, line 790-792:
*"Finally, the implementation of a surface saturation effect highlights the need for a more complex parameterization of desorption. In the future, further investigations using an equilibrium-type approach that depends on the mass ratio adsorbed/volatilised may bring key improvements."*

Specific Comments :

• Line 98-100: "the first two-box model allowing to consider the evolution of light intensity of each part along the day, as well as the volumes they occupy" Other partially illuminated models exist, including recent reports that use CFD for studies of HONO chemistry in partially illuminated spaces. It would be worthwhile to explicitly place the present model in a broader context.
>> This sentence has been modified as follows :
Line 99-101: *"This is one of the first models to consider the evolution of light intensity in each part through the day as well as the volumes that they occupy, without making use of CFD simulations (Won et al., 2019)."*
In the conclusion, the corresponding sentence now reads :
Line 774-775: *"It is also one of the first models to consider the variations of direct and indirect light throughout the day."*

• Line 103: "acquired in Martigues" Please be more specific.
>> Further details are now given in section 3.1 :
Line 365-366: *"Three experiments were conducted with anti-UV windows in an office room in a new building situated in the suburban area of Martigues (France), less than six months after its construction."*

• Line 138: "S_j_paint is the surface of paint in the box j [m-2]" The units suggest that this is a surface area-to-volume ratio, but the text suggests that it might be a surface area. Please clarify or correct.
>> This is indeed a typo which is now corrected.

• Line 158-159: "The complement of S_L_box to obtain the surface of the volume V_L_box is the same as the complement of S_S_box: : :" This wording is very confusing. Perhaps clarifying "surface area" and referring to the 'box' rather than 'the volume' would help.
>> "Volume" is now replaced by "box", which clarifies the sentence :
Line 161-163: *"The complement of S_L_box to obtain the total surface of the sunlit box is the same as the complement of S_S_box which is necessary to obtain the total surface of the shaded box."*

• Broader comment: Referring to various forms of "S" parameters as 'surfaces' appears in many locations. Please be more specific about the type of quantity that "S" denotes.
>> The "S" used for the Sutherland's constant has been replaced by "S_eta". The "S" exponent referring to the shaded box has been replaced by "G" :

Line 112-113: *"[...] 'L' denoting the sunlit box illuminated by direct light, and 'G' denoting the gloomy box illuminated indirectly by diffuse and reflected light (see Appendix A for a summary of the symbols used)."*

• Line 188-189: Similar efforts, but with ozone, have been made by Morrison et al. (Indoor Air, 2019). https://doi.org/10.1111/ina.12601

>> This reference has been added :

Line 192-193: *"Adsorption and decomposition reactions are modelled by combining transport to the boundary layer and surface adhesion (Mendez et al., 2015; Morrison et al., 2019), as detailed below."*

• Line 248-249: "deposition of the OH radical is considered negligible compared to its consumption by reactivity" Is this statement made in reference to the heterogeneous reactivity of OH or the homogeneous gas phase reactivity? Indoor multiphase reactivity by OH has been shown by Alwarda et al (Indoor Air, 2018).

>> This statement is made in reference to homogeneous gas-phase reactivity. OH radicals have a short lifetime so even if they can react with surfaces, this only affects the air parcels close to the walls, which is a small fraction of the total indoor volume.

Reference to the work on the OH heterogeneous chemistry is now included :

Line 251-254: *"Likewise, the deposition of the HO radical is considered as negligible compared with its consumption by homogeneous reactivity (Sarwar et al., 2002). There is evidence that low volatility species sorbed on surfaces can be subject to OH oxidation (Alwarda et al., 2018), but chemical variations of surface films caused by indoor oxidants are beyond the scope of this work."*

• Section 2.5: Have the authors considered a distance dependence to actinic flux? Dependence of actinic flux on wavelength, distance, and other variables have been characterized by Zhou et al. (Indoor Air, 2021) and Kowal et al. (Environ. Sci. Technol., 2017).

>> In the direct light, no distance dependence to actinic flux is considered. The distance with the source point of natural light (the sun) is so large that changes of a few meters within the sunlit area are not expected to affect the actinic flux (Kowal et al., 2017).

In the shaded area, light intensity is much weaker. With lower photolysis rates, photolytic reactions are expected to be poorly efficient in this area. Even if there is evidence that light intensity varies a lot with distance from its source (the sunlit area), such variations are expected to be negligible. The photolysis variations in the shade should be overwhelmed by the contributions of other reactions. Therefore, no distance dependence is considered at all :

Line 313-317: *"Whereas light intensity can be considered as homogeneous in the direct light whatever the distance from the window (Kowal et al., 2017), it decreases rapidly moving away from the direct sunlight (Gandolfo et al. 2016). The distribution of light intensity in the shaded volume is strongly location-specific, and thus hard to predict. However, light intensity in the shaded volume is much lower than the intensity of direct light; therefore the impact of the photolytic reactions occurring in the shaded volume is minor compared with that occurring in the sunlit volume."*

• Line 394-395: The definition of filtration factor here seems different than this reviewer's previous understanding. The filtration factor (or penetration factor, in Sarwar et al (2002)) describes the ability for a specific pollutant to penetrate the building envelope. It is not necessarily a measure of how airtight a building might be. It is related to the route and loss processes involved in transporting a particular pollutant from outdoors to indoors, and is independent of the air exchange rate (which is controlled in part by the airtight-ness of the building). NOx penetration factors have been measured by Zhao et al. (Environ. Sci. Technol., 2019; doi: 10.1021/acs.est.9b02920). These authors found that

the NO penetration factor was _1 and for NO2 it was closer to 0.7 for the same environment. Please clarify and correct the text. Note: This reviewer clearly understands that perhaps filtration or penetration factors have not been determined for the building being used in the present study, but citing values that are known for compounds in this study and that the filtration factor values are compound-specific are extremely important items for the reader to understand.

>> The text is now corrected as follows :

> Line 405-411: "*The building filtration factor controls the fluxes of outdoor pollutants that enter the room through the cracks and gaps of the building's structure. Its values derive from the routes available for transport and from the pollutants reactivity with the materials of the building's enclosure assembly, which can scavenge components that are infiltrating (Zhao et al., 2019). Its value is component-specific, and ranges from 0 (no intrusion) to 1. In the absence of measurements, it is omitted or taken as unity in most models (Sarwar et al., 2002; Carslaw, 2007; Mendez et al., 2015). For the present study, no filtration factor measurement was made. The filtration factor is thus considered as completely undetermined. For convenience, the same value is used for all compounds.*"

• Line 393-395: This statement about competitive adsorption of H2O and HONO may not be relevant. Our current understanding of indoor surfaces includes the possibility that multilayers may exist. Please refer the overall comment on the treatment of surface chemistry at the top of this review.

>> Indeed, this statement may not be relevant. We propose here a simple approach to parameterize desorption, based on previous work. This approach may be too simple, and may call for a more complex parameterization considering the mass ratio adsorbed/volatilised on the surface, as now discussed in section 7.

To make it clear that desorption is a model component that could be the subject of a specific work, the paragraph highlighted by this remark has been removed from the section "Model parameters" and is now placed in a new section 2.5 "Desorption rates".

• Line 405-406: "no experimental validation of this production ratio is available" This is not true, as it has been a topic of research for many years. Most saliently, the group of Finlayson-Pitts has studied this concept, and the authors have cited several of their papers. In addition, Grassian and co-workers have studied this problem.

>> The corresponding paragraph has been modified as follows :

> Line 417-425: "*Generally, it is assumed that HONO(ad) and HNO3(ad) are formed with equal yields (Febo and Perrino, 1991). Finlayson-Pitts et al. (2003) measured the yields of gas-phase HONO, NO and N2O, expressed relative to the measured losses of NO2 in the course of NO2 heterogeneous hydrolysis experiments in laboratory systems. The measured yield of HONO was less than 50% of the NO2 loss, but the NO yield was attributed to secondary reaction of the HONO formed by NO2 on surfaces. The sum of the yields of gas-phase HONO and secondary reaction products such as NO was close to 50%, but not exactly 50%. Furthermore, there is, to the author's knowledge, no available measurement of the HNO3 yield, since no HNO3 production is observed in the gas phase in the course of this type of experiments. By denoting NO2(ad) → beta_HNO3 HNO3(ad) + beta_HONO HONO(ad), small variations of beta_HNO3 and beta_HONO are considered, with the constraint that beta_HNO3 + beta_HONO = 1, to assure nitrogen conservation.*"

• Section 5.1.2: Filtration factor – The description of the filtration factor is largely correct, but some of the description may confuse the meaning of the filtration factor.

>> This description has been modified as follows :

> Line 539-541: "*It must be stressed that increasing f increases the intake of outdoor pollutants such as O3, NO and NO2, but not the losses caused by ventilation. These derive from the air exchange rate which remains unchanged.*"

• Line 523: "outdoor pollutants like O3 and NO2". Why isn't NO in this list? NO mixing ratios outdoors are often greater than NO2 during daytime in cities when NO2 photolysis is rapid.
>> NO has been added to the list.

• Line 533-534: Is it possible that the prevalence of Reaction S2 is overestimated?
>> In this model, the two possible significant NO sources are transport from outdoor and reaction S2. The value of kAER is constrained because measured experimentally, so the only way to increase transport from outdoor is to increase the filtration factor. Increasing this parameter leads to increase the concentrations of all the species present outdoors (see section 5.1.2). In particular, increasing the filtration factor leads to overestimate the NO2 concentration which is almost entirely controlled by this parameter. The filtration factor must be kept low enough to maintain a reasonable NO2 level, so that the only means to make the NO concentration match the measurements is to parameterize reaction S2 the way it is.
When using a balanced stoichiometry in reaction S1, it is not possible to simulate the steady state HONO increase that was observed experimentally, even by decreasing $k_{S_2}$.

• Line 534: "As NO increases, NO2 increases by equilibration through homogeneous chemistry." This statement is unclear. It may be correct, but it is not clear enough to understand what it is supposed to mean.
>> This sentence has been replaced by :
> Line 552-553: *"NO can be converted into NO2 by reacting with HO2, which increases the NO2 concentration."*

• Line 549-550: "The threshold value k_S1 = 0.003 s-1 is retained as it maximizes the NO2(ad) conversion." Why is this the most appropriate choice? Please further justify this decision. What impact does this decision have on other results?
>> In this model, it is difficult to make the simulated HONO concentration reach the HONO measurements. To meet this goal, the production of HONO(ad) must be maximized, which amounts to saying that the conversion of NO2(ad) must be as efficient as possible. As a result, there is less NO2(ad) on surfaces, thus less NO2 desorption. This does not seem to have major consequences, considering that NO2 is essentially brought by transport from outdoors.

• Section 5.3.1: What happens if HONO and HNO3 dissociate?
>> If HONO(ad) and HNO3(ad) dissociate, the production of NO(ad) by reaction S2 is decreased. In that case, a good match between measured and simulated NO concentrations could be preserved by increasing the NO desorption constant.

• Line 633: Please comment on how the uptake coefficient of HONO compares with the literature.
>> To the authors' knowledge, there is a lack of measurements of HONO uptakes on indoor surfaces in the literature. El Zein and Bedjanian (2012) found that the reactive uptake of HONO under dark conditions was independent of HONO concentration and weakly dependent on temperature, but strongly impacted by humidity. The following sentence has been added :
> Line 655-658: *"By comparison, considering an average humidity of 45%, the relationship between humidity and gamma_HONO measured on TiO2 surface found by El Zein and Bedjanian (2012) under dark conditions gives gamma_HONO = 1.6 × 10−6 , which is a value higher than the one determined here. However, we recall that the same uptake value is used for all the materials constituting the room (walls, floor, window), which complicates the analogy."*

• Line 650-653: It is somewhat alarming to see that the parameters within the model must be altered in response to the reference experiments. The broader issue brought forth by Section 6 is discussed in the overall comments of this review and has been addressed by the authors.

>> The article presents three types of parameters optimization, including an attempt to use common values for the three experiments (Set 3). This attempt is not perfect, but leads to acceptable results. In section 6, we focus on a particular experiment, this is why we opt for the set of parameters that optimizes the correspondence between simulations and measurements (Set 1). Uncertainty on the model parameters is discussed in Section 7.

• Line 702: Nitric acid is most reactive with other radical species or reducing agents. It is not terribly reactive in the indoor environment – not close at all to the reactivity of OH. The small deposition velocity for NO comes from its slow reactivity with surface-bound materials. If there is more information about the reactivity of NO indoors, please cite.

>> We observe that the gas-phase NO concentration is barely influenced by variations of NO uptake, which means that deposition has a negligible contribution to the gas-phase NO concentration, compared to other sink terms. The two other possible sinks for NO are reactivity and ventilation. Considering that ventilation is not particularly efficient in these experiments, we can conclude that reactivity is the major sink for NO. Due to the very low impact of deposition on the gas-phase NO concentration, we can set it to a quasi-null value, which is not supposed to be its genuine value, but an apparent value. The text has been modified as follows :

> Line 732-735: *"Similarly to the OH radical, NO is likely too reactive to be affected by deposition. When deposition has a negligible contribution compared with homogeneous reactivity, the influence of the species uptake coefficient on gas-phase concentration can be considered to be null, whatever its genuine value may be. In that case, the uptake coefficient can be set to zero, as an apparent uptake value."*

**Anonymous Referee #2**

Interesting paper that presents a model that considers homogeneous and heterogeneous chemistry. Some remarks and suggestions :

• The English must be improved in all parts of the paper,

>> The English is now proofread and corrected by a professional editing service.

• page 1, line 8, "The model succeeds in simulating correctly", what you mean by correctly, please to precise, e.g., give error values comparing simulations and experiments

>> The error value is not the same for the species simulated, and the uncertainty is not the same for all the measured species. Giving the same reference for the four species would be deceitful. The word "correctly" has been removed from the sentence, as it is indeed too subjective.

• page 4, line 107 suggestion to replace "master equation" mass balance equation "

>> This title has been modified as suggested.

• page 4, line 110, I suggest presenting a simple schema that clearly shows the domains of L, S, the boundary of domains, and all flows and mass fluxes, this consideration will help to understand more the model

>> An example of schema is now included.

• page 5, line 121/Eq2, could you explain why you multiply miS by VboxL/VboxS / same question for line 123/Eq3

>> The mass of species i introduced in box L from box S is proportional to the mass concentration in box S, defined as miS/VboxS, applied to the volume VboxL. Reciprocally, the mass of species i introduced in box S from box L is proportional to the mass concentration in box L, defined as miL/VboxL, applied to the volume VboxS.

• page 5, line 133, ... in these experiments..., which experiments? please precise
>> The text is now : "the experiments of the considered campaign".

• Page 5, ... emission of the building itself... not clear which emission? Indoor emissions due to activities/appliances/furnitures?
>> The text now specifies : "[...] released by the building materials, furniture and appliances of the neighbouring rooms".

• page 6, line 149-line 156, must present a schema that shows an example of surface projections, surface boundaries, beams with window the window position considered
>> The schema that is now included (see above) helps understand how light irradiates from the window, but the technical schemes presenting the surface projections of the sunlit volume are the very topic of a specific manuscript (Tlili et al., in prep).

• page 6, line 153, line 155, are these equations valid for each season? Each window position, surface and characteristics? please to highlight the limit of these equations,
>> Indeed, these equations are completely site's specific. The following sentence has been added :
    Line 158-160: "We stress that Eqs. (11) and (12) are valid only for the geometry of the room where the measurements were acquired (building orientation, window position), and for the period during which the campaign was performed (end of October)."

• page 12, line 339, what parameters are assigned for VOCs, not clear, VOCs concentrations?
>> In this work, the VOCs concentrations are constrained to their measured values, so all the parameterizations of emission and deposition are not considered for VOCs.  It may be somewhat confusing, indeed, to present all the parameters necessary to compute the VOCs deposition velocities, whereas these velocities are not used in this paper. These elements have been removed from table 2, and the text at the end of section 2.4.1 has been modified accordingly.

• page 12, line 341, is slower? it seems it is lower, not clear
>> Indeed, "slower" is changed for "lower".

• page 14, line 388, how the other uptake values can be adjusted by a user? From values in the literature?
>> The user can adjust the uptake coefficient based on values taken from the literature, but the difficulty is that the same uptake value is used for every surfaces, including window (glass), floor and ceiling. It could be possible to distinguish the uptakes of the surfaces present, but this calls for more detailed experimental characterization of the materials commonly encountered in indoor environments in various conditions of temperature, light and humidity.

• page 25, line 734, suggestion to replace "replicate" by "to predict"
>> « replicate » has been replaced by « simulate » in this sentence.

• page 25, line 735, please to review this phrase "For the first time, O₃, HONO and NOx species are simulated all at once and compared to experimental records acquired in a real room". is it the first study that estimates experimentally O₃, HONO and NOx concentrations in a real environment?
>> No, similar indoor VOCs, O₃, NO, NO₂ and HONO measurements where performed during the SURFin campaign (Alvarez et al. 2013). But to the author's knowledge, it is the first time these species are modelled together, with a comparison to measurements acquired in a room.

• page 25, line 737 and line 738, please review this phrase "It is also the first two-box model allowing to consider the variations of direct and indirect light throughout the day".please to more explicit about this consideration by making the link with the study

>> The paragraph is modified as follows :

Line 774-778: *"It is also one of the first models to consider the variations of direct and indirect light throughout the day. This feature is of particular importance in studying the impact of photolytic processes on indoor chemistry. After developing and testing the model in the absence of these processes in the present work, it can now be used to simulate data obtained with UVs-transparent windows allowing photochemistry to occur."*